# Building Deep Equivariant Capsule Networks

**Sai Raam Venkataraman, S. Balasubramanian & R. Raghunatha Sarma**
Department of Mathematics and Computer Science
Sri Sathya Sai Institute of Higher Learning
`{vsairaam,sbalasubramanian,rraghunathasarma}@sssihl.edu.in`

## Abstract

Capsule networks are constrained by the parameter-expensive nature of their layers, and the general lack of provable equivariance guarantees. We present a variation of capsule networks that aims to remedy this. We identify that learning all pair-wise part-whole relationships between capsules of successive layers is inefficient. Further, we also realise that the choice of prediction networks and the routing mechanism are both key to equivariance. Based on these, we propose an alternative framework for capsule networks that learns to projectively encode the manifold of pose-variations, termed the space-of-variation (SOV), for every capsule-type of each layer. This is done using a trainable, equivariant function defined over a grid of group-transformations. Thus, the prediction-phase of routing involves projection into the SOV of a deeper capsule using the corresponding function. As a specific instantiation of this idea, and also in order to reap the benefits of increased parameter-sharing, we use type-homogeneous group-equivariant convolutions of shallower capsules in this phase. We also introduce an equivariant routing mechanism based on degree-centrality. We show that this particular instance of our general model is equivariant, and hence preserves the compositional representation of an input under transformations. We conduct several experiments on standard object-classification datasets that showcase the increased transformation-robustness, as well as general performance, of our model to several capsule baselines.

## 1 Introduction

The hierarchical component-structure of visual objects motivates their description as instances of class-dependent spatial grammars. The production-rules of such grammars specify this structure by laying out valid type-combinations for components of an object, their inter-geometry, as well as the behaviour of these with respect to transformations on the input. A system that aims to truly understand a visual scene must accurately learn such grammars for all constituent objects - in effect, learning their aggregational structures. One means of doing so is to have the internal representation of a model serve as a component-parsing of an input across several semantic resolutions. Further, in order to mimic latent compositionalities in objects, such a representation must be reflective of detected strengths of possible spatial relationships. A natural structure for such a representation is a parse-tree whose nodes denote components, and whose weighted parent-child edges denote the strengths of detected aggregational relationships.

Capsule networks (Hinton et al., 2011), (Sabour et al., 2017) are a family of deep neural networks that aim to build such distributed, spatially-aware representations in a multi-class setting. Each layer of a capsule network represents and detects instances of a set of components (of a visual scene) at a particular semantic resolution. It does this by using vector-valued activations, termed 'capsules'. Each capsule is meant to be interpreted as being representative of a set of generalised pose-coordinates for a visual object. Each layer consists of capsules of several types that may be instantiated at all spatial locations depending on the nature of the image. Thus, given an image, a capsule network provides a description of its components at various 'levels' of semantics. In order that this distributed representation across layers be an accurate component-parsing of a visual scene, and capture meaningful and inherent spatial relationships, deeper capsules are constructed from shallower capsules using a mechanism that combines backpropagation-based learning, and consensus-based heuristics.

Briefly, the mechanism of creating deeper capsules from a set of shallower capsules is as follows. Each deeper capsule of a particular type receives a set of predictions for its pose from a local pool of shallower capsules. This happens by using a set of trainable neural networks that the shallower capsules are given as input into. These networks can be interpreted as aiming to capture possible part-whole relationships between the corresponding deeper and shallower capsules. The predictions thus obtained are then combined in a manner that ensures that the result reflects agreement among them. This is so that capsules are activated only when their component-capsules are in the right spatial relationship to form an instance of the object-type it represents. The agreement-based aggregation described just now is termed 'routing'. Multiple routing algorithms exist, for example dynamic routing (Sabour et al., 2017), EM-routing (Hinton et al., 2018), SVD-based routing (Bahadori, 2018), and routing based on a clustering-like objective function (Wang & Liu, 2018).

Based on their explicit learning of compositional structures, capsule networks can be seen as an alternative (to CNNs) for better learning of compositional representations. Indeed, CNN-based models do not have an inherent mechanism to explicitly learn or use spatial relationships in a visual scene. Further, the common use of layers that enforce local transformation-invariance, such as pooling, further limit their ability to accurately detect compositional structures by allowing for relaxations in otherwise strict spatial relations (Hinton et al., 2011). Thus, despite some manner of hierarchical learning - as seen in their layers capturing simpler to more complex features as a function of depth - CNNs do not form the ideal representational model we seek. It is our belief that capsule-based models may serve us better in this regard.

This much said, research in capsule networks is still in its infancy, and several issues have to be overcome before capsule networks can become universally applicable like CNNs. We focus on two of these that we consider as fundamental to building better capsule network models. First, most capsule-network models, in their current form, do not scale well to deep architectures. A significant factor is the fact that all pair-wise relationships between capsules of two layers (upto a local pool) are explicitly modelled by a unique neural network. Thus, for a 'convolutional capsule' layer - the number of trainable neural networks depends on the product of the spatial extent of the windowing and the product of the number of capsule-types of each the two layers. We argue that this design is not only expensive, but also inefficient. Given two successive capsule-layers, not all pairs of capsule-types have significant relationships. This is due to them either representing object-components that are part of different classes, or being just incompatible in compositional structures. The consequences of this inefficiency go beyond poor scalability. For example, due to the large number of prediction-networks in this design, only simple functions - often just matrices - are used to model part-whole relationships. While building deep capsule networks, such a linear inductive bias can be inaccurate in layers where complex objects are represented. Thus, for the purpose of building deeper architectures, as well as more expressive layers, this inefficiency in the prediction phase must be handled.

The second issue with capsule networks is more theoretical, but nonetheless has implications in practice. This is the lack, in general, of theoretical guarantees on equivariance. Most capsule networks only use intuitive heuristics to learn transformation-robust spatial relations among components. This is acceptable, but not ideal. A capsule network model that can detect compositionalities in a provably-invariant manner are more useful, and more in line with the basic motivations for capsules.

Both of the above issues are remedied in the following description of our model. First, instead of learning pair-wise relationships among capsules, we learn to projectively encode a description of each capsule-type for every layer. This we do by associating each capsule-type with a vector-valued function, given by a trainable neural network. This network assumes the role of the prediction mechanism in capsule networks. We interpret the role of this network as a means of encoding the manifold of legal pose-variations for its associated capsule-type. It is expected that, given proper training, shallower capsules that have no relationship with a particular capsule-type will project themselves to a vector of low activation (for example, 2-norm), when input to the corresponding network. As an aside, it is this mechanism that gives the name to our model. We term this manifold the 'space-of-variation' of a capsule-type. Since, we attempt to learn such spaces at each layer, we name our model 'space-of-variation' networks (SOVNET). In this design, the number of trainable networks for a given layer depend on the number of capsule-types of that layer.

As mentioned earlier, the choice of prediction networks and routing algorithm is important to having guarantees on learning transformation-invariant compositional relationships. Thus, in order to ensure equivariance, which we show is sufficient for the above, we use group-equivariant convolutions (GCNN) (Cohen & Welling, 2016) in the prediction phase. Thus, shallower capsules of a fixed type are input to a GCNN associated with a deeper capsule-type to obtain predictions for it. Apart from ensuring equivariance to transformations, GCNNs also allow for greater parameter-sharing (across a set of transformations), resulting in greater awareness of local object-structures. We argue that this could potentially improve the quality of predictions when compared to isolated predictions made by convolutional capsule layers, such as those of (Hinton et al., 2018).

The last contribution of this paper is an equivariant degree-centrality based routing algorithm. The main idea of this method is to treat each prediction for a capsule as a vertex of a graph, whose weighted edges are given by a similarity measure on the predictions themselves. Our method uses the softmaxed values of the degree scores of the affinity matrix of this graph as a set of weights for aggregating predictions. The key idea being that predictions that agree with a majority of other predictions for the same capsule get a larger weight - following the principle of routing-by-agreement. While this method is only heuristic in the sense of optimality, it is provably equivariant and preserves the capsule-decomposition of an input. We summarise the contributions of this paper in the following:

1. A general framework for a scalable capsule-network model.

2. A particular instantiation of this model that uses equivariant convolutions, and an equivariant, degree-centrality-based routing algorithm.

3. A graph-based framework for studying the representation of a capsule network, and the proof of the sufficiency of equivariance for the (qualified) preservation of this representation under transformations of the input.

4. A set of proof-of-concept, evaluative experiments on affinely transformed variations of MNIST, FASHIONMNIST, and CIFAR10, as well as separate experiments on KMNIST and SVHN that showcase the superior adapatability of SOVNET architectures to train and test-time geometric perturbations of the data, as well as their general performance.

## 2    SOVNET, EQUIVARIANCE, AND COMPOSITIONALITY

We begin with essential definitions for a layer of SOVNET, and the properties we wish to guarantee. Given a group $(G, \circ)$, we formally describe the $l^{th}$ layer of a SOVNET architecture as the set of function-tuples $\{(f_i^l, a_i^l) : 0 \leq i \leq N_l - 1; f_i^l : G \to \mathbb{R}^{d^l}; a_i^l : G \to [0, 1]\}$. Here, $N_l$ denotes the number of capsule-types at the $l^{th}$ layer, $f_i^l$ is a functional description of the $d^l$-dimensional pose-vectors of instances of the $i^{th}$ capsule-type, and $a_i^l$ is a functional description of the corresponding activations.

We model each capsule-type as a function over a group of transformations so as to allow for formal guarantees on transformation-equivariance. Thus, we also model images as function from a group to a representation-space. The main assumption being that the translation-group is a subgroup of the group in question. This is similar in approach to (Cohen & Welling, 2016). We wish for each capsule-type, both pose and activation-wise, to display equivariance. We present a formal definition of this notion.

Consider a group $(G, \circ)$ and vector spaces $V, W$. Let $T$ and $T'$ be two group-representations for elements of $G$ over $V$ and $W$, respectively. $\Phi \colon V \to W$ is said to be equivariant with respect to $T$ and $T'$ if $\forall g \in G, \forall x \in V, \Phi(T_g x) = T'_g \Phi(x)$.

This definition translates to a preservation on transformations in the input-space to the output-space - something that allows no loss of information in compositional structures. As in (Cohen & Welling, 2016), we restrict the notion of equivariance in our model by using the operator $L_g$ in place of the group-representation. $L_g$ is given by $[L_g f](x) = f(g^{-1}x)$. Thus, if $\otimes$ denotes an operation between two functions, we require $([L_g f] \otimes \Psi)(x) = [L_g(f \otimes \Psi)](x)$. The operator $\otimes$ describes the change in representation space, and is dependent on the nature of the deep learning model. In the case of

capsule networks (and SOVNET), this change is given by routing among capsules as described in subsection 2.1.

## 2.1 SOVNET LAYER

We define the capsule-types of a particular layer as an output of an agreement-based aggregation of predictions made by the preceding layer. A recursive application of this definition is enough to define a SOVNET architecture, given an initial set of capsules. A means of obtaining this initial set is given in section 3. We provide a general framework for the summation-based family of routing procedures in Algorithm 1.

---

**Algorithm 1** A general summation-based routing algorithm for SOVNET.

---

**Input**: $\{(f_i^l, a_i^l) | i \in \{0, ..., N_l - 1\}, f_i^l : G \to \mathbb{R}^{d^l}, a_i^l : G \to [0, 1]\}$
**Output**: $\{(f_j^{l+1}, a_j^{l+1}) | j \in \{0, ..., N_{l+1} - 1\}, f_j^{l+1} : G \to \mathbb{R}^{d^{l+1}}, a_j^{l+1} : G \to [0, 1]\}$
**Trainable Functions**: $(\Psi_j^{l+1}, \cdot)$ - projection networks that use operator $\cdot$

$S_{ij}^{l+1}(g) = ((f_i^l, a_i^l) \cdot \Psi_j^{l+1})(g) \, \forall \, i, j, \forall g \in G$
$(c_{0j}^{l+1}(g), ..., c_{N_l-1 j}^{l+1}(g)) = GetWeights(S_{0j}^{l+1}(g), ..., S_{N_l-1 j}^{l+1}(g)) \, \forall \, j, \forall g \in G$
$f_j^{l+1}(g) = \sum_{i=1}^{N_l-1} c_{i,j}^{l+1}(g) S_{ij}^{l+1}(g) \, \forall \, j, \forall g \in G$
$a_j^{l+1}(g) = Agreement(f_j^{l+1}(g), S_{0j}^{l+1}(g), ..., S_{N_l-1 j}^{l+1}(g)) \, \forall \, j$

---

The weighted-sum family of routing algorithms builds deeper capsules using a weighted sum of predictions made for them by shallower capsules. To ensure that the predictions are combined in a meaningful manner, different methods can be used to obtain the weights. The role of the function $GetWeights$ is to represent any such mechanism. The activation of a capsule, representative of the probability of existence of the object it represents, is determined by the extent of the consensus among its predictions. This is based on the routing-by-agreement principle of capsule networks. The $Agreement$ function represents any means of evaluating such consensus.

We instantiate the above algorithm to a specific model, as given in Algorithm 2. In this model, the $\Psi_j^l$ are group-equivariant convolutional filters, and the operator $\cdot$ is the corresponding group-equivariant correlation operator $\star$. The weights $c_{ij}^{l+1}(g)$ are, in this routing method, the softmaxed degree-scores of the affinities among predictions for the same deeper capsule. Further, like in dynamic routing (Sabour et al., 2017), we also assume that the activation of a capsule is given by its 2-norm. To ensure that this value is in $[0, 1]$, we use the 'squash' function of dynamic routing. Thus, we do not mention it explicitly. Note that we have used the subscript notation to also denote that a variable is part of a vector, for example $S_{ijp}^{l+1}(g)$ denotes the $p^{th}$ element of the $d^{l+1}$-dimensional vector $S_{ij}^{l+1}(g)$. This new routing algorithm is meant to serve as an alternative to existing iterative routing strategies such as dynamic routing. An important strength of our method being that there is no hyperparameter, like that of the number of iterations in dynamic routing or EM routing.

## 2.2 EQUIVARIANCE, COMPOSITIONALITY AND SOVNET

The SOVNET layer we introduced in Algorithm 2 is group-equivariant with respect to the group action $L_g$, where $g \in G$ - the set of transformations over which the group-convolution is defined. For notational convenience, we define $\otimes$ to be an operator that encapsulates the degree-routing procedure with prediction networks $\Psi_j^{l+1}$. Thus, the $j^{th}$ capsule-type of the $l+1^{th}$ layer is functionally depicted as $f_j^{l+1} = (F^l \otimes \Psi_j^{l+1})$, where $F^l = (f_0^l, ..., f_{N_l-1}^l)$. The formal statement of this result is given below; the proof is presented in the appendix.

**Theorem 2.1.** *The SOVNET layer defined in Algorithm 2, and denoted by the operator $\otimes$ as given above, satisfies $([L_g F^l] \otimes \Psi_j^{l+1}) = (L_g [F^l \otimes \Psi_j^{l+1}])$, where g belongs to the underlying group of the equivariant convolution.*

*Proof.* The proof is given in the appendix. $\square$

---

**Algorithm 2** The degree-centrality based routing algorithm for SOVNET.

---

**Input**: $\{f_i^l | i \in \{0, ..., N_l - 1\}, f_i^l : G \to \mathbb{R}^{d^l}\}$

**Output**: $\{f_j^{l+1} | j \in \{0, ..., N_{l+1} - 1\}, f_j^{l+1} : G \to \mathbb{R}^{d^{l+1}}\}$

**Trainable Functions**: $(\Psi_j^{l+1}, \star), 0 \le j \le N_{l+1} - 1$, - a set of $d^{l+1}$ group-equivariant convolutional filters (per capsule-type) that use the group-equivariant correlation operator $\star$

$\quad S_{ijp}^{l+1}(g) = (f_i^l \star \Psi_j^{l+1,p})(g) = \sum_{h \in G} \sum_{k=0}^{d^l-1} f_{ik}^l(h) \Psi_k^{l+1,p}(g^{-1} \circ h); p \in \{0, ..., d^{l+1} - 1\}$

$\quad (c_{0j}^{l+1}(g), ..., c_{N_l-1j}^{l+1}(g)) = DegreeScore(S_{0j}^{l+1}(g), ..., S_{N_l-1j}^{l+1}(g)); \forall 0 \le j \le N_{l+1} - 1, \forall\, g$

$\quad f_j^{l+1}(g) = \sum_{i=0}^{N_l-1} c_{ij}^{l+1}(g) S_{ij}^{l+1}(g) \,\forall\, 0 \le j \le N_{l+1} - 1, \forall g \in G$

$\quad f_j^{l+1}(g) = Squash(f_j^{l+1}(g)) = \frac{\|f_j^{l+1}(g)\|_2}{1+\|f_j^{l+1}(g)\|_2^2} \, f_j^{l+1}(g); \forall\, 0 \le j \le N_l - 1, \forall\, g \in G$

$\quad$ **procedure** DEGREESCORE($S_{0j}^{l+1}(g), ..., S_{N_l-1j}^{l+1}(g)$)

$A_{ik}^j(g) = \frac{S_{ij}^{l+1}(g) \cdot S_{kj}^{l+1}(g)}{\|S_{ij}^{l+1}(g)\|_2 \cdot \|S_{kj}^{l+1}(g)\|_2}; 0 \le i, k \le N_l - 1$

$Degree_i^j(g) = \sum_{k=0}^{N_l-1}(A_{ik}^j(g)); 0 \le i \le N_l - 1$

$c_{ij}(g) = \frac{\exp(Degree_i^j(g))}{\sum_{k=0}^{N_l-1} \exp(Degree_k^j(g))}; 0 \le i \le N_l - 1$

$\quad$ **return** $c_{ij}(g) \,\forall 0 \le i \le N_l - 1$

---

Equivariance is widely considered a desirable inductive bias for a variety of reasons. First, equivariance mirrors natural label-invariance under transformations. Second, it lends predictability to the output of a network under (fixed) transformations of the input. These, of course, lead to a greater robustness in handling transformations of the data. We aim at adding to this list by showing that equivariance guarantees the preservation of detected compositionalities in a SOVNET architecture. This is of course quite unsurprising, and has been a significant undercurrent of the capsule-network idea. Our work completes this intuition with a formal result.

We begin by first defining the notion of a capsule-decomposition graph. This graph is formed from the activations and the routing weights of a SOVNET. Specifically, given an input to a SOVNET model, each capsule of every type is a vertex in this graph. We construct an edge between capsules that are connected by routing, with the direction from the shallower capsule to the deeper capsule. Each of these edges are weighted by the corresponding routing coefficient. Capsules not related to each other by routing are not connected by an edge. This graph is a direct formalisation of the various detected compositionalities with their strengths.

What should the ideal behaviour of this graph be under the change-of-viewpoint of an input? The answer to this lies in the expected behaviour of natural compositionalities. Thus, while the pose of objects, and their components, is changed under transformations of the input, the relative geometry is constant. Thus, it is desirable that the capsule-decomposition graphs of a particular input (and its transformed variations) be isomorphic to each other. We show that a SOVNET model that is equivariant with respect to a set of transformations satisfies the above property for that set. A more formal description of the capsule-decomposition graph, and the statement for the above theorem are given below.

Consider an $L$-layer SOVNET model, whose routing procedure belongs to the family of methods given by Algorithm 1. Let us consider a fixed input $x : G \to \mathbb{R}^c$. We define the capsule-decomposition graph of such a model, for this input $x$, as $\mathbb{G}(x) = (V(x), E(x))$. Here, $V(x)$ and $E(x)$ denote the vertex-set and the edge-set, respectively. $V(x) = \{\tilde{f}_i^l(g) : 0 \le i \le N_l - 1, 0 \le l \le L - 1, g \in G\}$, where $\tilde{f}_i^l(g) = (g, i, f_i^l(g), a_i^l(g))$, $g \in G, 0 \le i\, N_l - 1$. $E(x) = \{(\tilde{f}_i^l(g_1), \tilde{f}_j^{l+1}(g_2), c_{ij}^{l+1}(g_2)) : g_1 \in Pool_j^{l+1}(g_2)\}$. $Pool_j^{l+1}(g_2)$ denotes the pool of grid-positions at layer $l$ that route to the deeper capsule of type $j$ of layer $l + 1$ at $g_2$. A more formal definition is given the appendix. We also use the notation $L_h \tilde{f}_i^l(g)$ to denote $(h^{-1} \circ g, i, f_i^l(h^{-1} \circ g), a_i^l(h^{-1} \circ g))$.

**Theorem 2.2.** *Consider an $L$-layer SOVNET whose activations are routed according to a procedure belonging to the family given by Algorithm 1. Further, assume that this routing procedure is*

*equivariant with respect to the group G. Then, given an input $x$ and $\forall g \in G$, $\mathbb{G}(x)$ and $\mathbb{G}([L_g x])$ are isomorphic.*

*Proof.* The proof is given in the appendix. □

Based on above theorem, and the fact that degree-centrality based routing is equivariant, the above result applies to SOVNET models that use Algorithm 2 .

## 3 EXPERIMENTS AND RESULTS

This section presents a description of the experiments we performed. We conducted two sets of experiments; the first to compare SOVNET architectures to other capsule network baselines with respect to transformation robustness on classification, and the second to compare SOVNET to certain capsule as well as convolutional baselines based on classification performance. Before we present the details of these experiments, we briefly describe some details of the SOVNET architecture we used. We only present an outline - the complete details, both architecture-wise and about the training, can be found in the anonymised github repository `https://github.com/sairaamVenkatraman/ SOVNET`.

The first detail of the architecture pertains to the construction of the first layer of capsules. While many approaches are possible, we used the following methodology that is similar in spirit to other capsule network models. The first layer of the SOVNET architectures we constructed use a modified residual block that uses the SELU activation, along with group-equivariant convolutions. This is so as to allow a meaningful set of equivariant feature maps to be used for the creation of the first set of capsules. Intuition and some literature, for example Rosario et al. (2019), suggest that the construction of primary capsules plays a significant role in the performance of the capsule network. Thus, it is necessary to build a sufficiently expressive layer that yields the first set of meaningful capsule-activations. To this end, each capsule-type in the primary capsule layer is associated with a group-convolution layer followed by a modified residual block. The convolutional feature-maps from the preceding layer passes through each of these sub-networks to yield the primary capsules. No routing is performed in this layer.

We now describe the SOVNET blocks. Since the design of SOVNET significantly reduces the number of prediction networks, and thereby the number of trainable parameters, we are able to build architectures whose each layer uses more expressive prediction mechanisms than a simple matrix. Specifically, each hidden layer of the SOVNET architectures we consider uses a (group-equivariant) modified residual block as the prediction mechanism. We use a SOVNET architecture that uses 5 hidden layers for MNIST, FashionMNIST, KMNIST, and SVHN, and a model that uses 6 hidden layers for CIFAR-10. Unlike DeepCaps - another capsule network whose predictions use (regular) convolution, each of the hidden layers of our SOVNET models use degree-routing. The hidden layers of DeepCaps (excepting the last), in contrast, are not strictly capsule-based - being just convolutions whose outputs are reshaped to a capsule-form.

The output capsule-layer of SOVNET is designed similar to the hidden capsule-layers, with the difference that the prediction-mechanism is a group-convolutional implementation of a fully-connected layer. In order to make a prediction for the class of an input, the maximum across the rotational (and reflectional) positions of the two-norm of the capsule-activations of this layer are taken for each class-type. This is an equivariant operation, as it corresponds to the subgroup-pooling of Cohen & Welling (2016). The predictions that this layer yields is the type of the capsule with the maximum 2-norm.

In order to guarantee the robustness to translations and rotations, we used the p4-convolutions (Cohen & Welling, 2016) for the prediction mechanism in all the networks used in the first set of experiments. For the second set, we used the p4m-convolution (Cohen & Welling, 2016), that is equivariant to rotations, translations and reflections - for greater ability to learn from augmentations. The architectures, however are identical but for this difference.

As in (Sabour et al., 2017), we used a margin loss and a regularising reconstruction loss to train the networks. The positive and negative margins for half of the training epochs were set to 0.9 and 0.1, respectively. Further, the negative margin-loss was weighted by 0.5, as in (Sabour et al., 2017). These values were used for the first half of the training epochs. In order to facilitate better predictions, these values were changed to 0.95, 0.05, and 0.8, respectively for the second half of the training. We adopt this from (Rajasegaran et al., 2019). The reconstruction loss was computed by masking the incorrect classes, and by feeding the 'true' class-capsule to a series of transposed convolutions to reconstruct the image. The mean square loss was computed for the reconstruction and original image. The main idea being that this loss guides the capsule network to build meaningful capsules. This loss was weighed by 0.0005 as in (Sabour et al., 2017). We used the Adam optimiser and an exponential learning rate scheduler that reduced the learning rate by a factor of 0.9 each epoch.

With this outline of the architecture and details of the training, we now describe the first set of experiments we conducted on SOVNET. The preservation of detected compositionalities under transformations in SOVNET leads us to the expectation that SOVNET models, when properly trained, will display greater robustness to changes in viewpoint of the input. Apart from handling test-time transformations, as is the commonly held notion of transformation robustness, a robust model must also effectively learn from train-time perturbations of the data. Based on these ideas, we designed a set of experiments that compare SOVNET architectures to other capsule networks on their ability to handle train and test-time affine transformations of the data.

Specifically, we perform experiments on MNIST (LeCun & Cortes, 2010), FashionMNIST (Xiao et al., 2017), and CIFAR-10 (Krizhevsky & Hinton, 2009). For each of these datasets, we created 5 variations of the train and test-splits by randomly transforming data according to the extents of the transformations given in Table 1. We train a given model on each transformed version of the training-split, and test each model on each of the versions of the test-split. Thus we obtain, for a single model, 25 accuracies per dataset - each corresponding to a pair of train and test-splits. There is a single modification to these transformations for the case of CIFAR-10. In order to compare SOVNET against the closest competitor DeepCaps, we use their strategy of first resizing CIFAR-10 images to 64×64, followed by translations and rotations.

We tested SOVNET against four capsule network baselines, namely Capsnet (Sabour et al., 2017), EMcaps (Hinton et al., 2018), DeepCaps Rajasegaran et al. (2019), and GCaps (Lenssen et al., 2018). The results of these experiments are given in Tables 2 to 4. In the majority of the cases, SOVNET obtains the highest accuracy - showing that it is more robust to transformations of the data. Note that we had to conduct these experiments as such a robustness study was not done in the original papers for the baselines. We used, and modified, code from the following github sources for the implementation of the baselines: (Li, 2019) for CAPSNET; (Yang, 2019) for EMCAPS; (Rajasegaran, 2019) and (HopefulRational, 2019) for DeepCaps, and (Lenssen, 2019) for GCaps. We also tested against a group-equivariant convolution network (GCNN).

The second set of experiments we conducted, tested SOVNET against several capsule as well as convolutional baselines. We trained and tested SOVNET on KMNIST (Clanuwat et al., 2018) and SVHN (Netzer et al., 2011). With fairly standard augmentation - mild translations (and resizing for SVHN to 64×64) - the SOVNET architecture with p4m-convolutions was able to achieve on-par, or above, comparative performance. The results of this experiment are in Table 5. In order to compare the performance of SOVNET architectures against more sophisticated CNN-baselines, we also trained ResNet-18, ResNet-34 on the most extreme transformation - translation by up to $\pm$ 2 pixels, and rotation by up to $\pm$ 180°. The results of these experiments are presented in the appendix.

## 4   Discussion and related work

A number of insights can be drawn from an observation of the accuracies obtained from the experiments. First, the most obvious, is that SOVNET is significantly more robust to train and test-time geometric transformations of the input. Indeed, SOVNET learns to use even extreme transformations of the training data and generalises better to test-time transformations in a majority of the cases. However, in certain splits, some baselines perform better than SOVNET. These cases are briefly discussed below.

Table 1: List of the extents for the affine transformations.

| S.no. | Translational extent | Rotational extent |
|---|---|---|
| 1 | 0 pixels | 0° |
| 2 | 2 pixels | 30° |
| 3 | 2 pixels | 60° |
| 4 | 2 pixels | 90° |
| 5 | 2 pixels | 180° |

Table 2: Experiments on MNIST.

| Results on Training on Untransformed MNIST | | | | | Results on Training on MNIST Transformed by (2,30°) | | | | |
|---|---|---|---|---|---|---|---|---|---|
| Method | (0,0°) | (2,30°) | (2,60°) | (2,90°) | (2,180°) | Method | (0,0°) | (2,30°) | (2,60°) | (2,90°) | (2,180°) |
| Capsnet | 99.35% | 91.57% | 72.10% | 55.27% | 42.58% | Capsnet | 99.60% | 99.39% | 95.65% | 79.53% | 59.58% |
| EMcaps | 99.09% | 92.23% | 72.83% | 56.66% | 42.95% | EMcaps | 99.36% | 99.03% | 94.91% | 79.12% | 59.03% |
| G-Caps | 97.83% | 82.59% | 66.27% | 56.63% | **54.52%** | G-Caps | 98.12% | 96.17% | 90.87% | 81.34% | **77.13%** |
| DeepCaps | 99.56% | 94.61% | 74.44% | 57.24% | 45.43% | DeepCaps | 99.62% | 99.57% | 97.50% | 84.16% | 62.75% |
| GCNN | 99.61% | 93.96% | 75.53% | 58.91% | 46.07&  | GCNN | 99.67% | 99.46% | 97.11% | 84.5% | 63.74% |
| SOVNET | **99.68%** | **96.15%** | **80.53%** | **64.55%** | 51.02% | SOVNET | **99.77%** | **99.70%** | **98.86%** | **90.63%** | 69.26% |

| Results on Training on MNIST Transformed by (2,60°) | | | | | Results on Training on MNIST Transformed by (2,90°) | | | | |
|---|---|---|---|---|---|---|---|---|---|
| Method | (0,0°) | (2,30°) | (2,60°) | (2,90°) | (2,180°) | Method | (0,0°) | (2,30°) | (2,60°) | (2,90°) | (2,180°) |
| Capsnet | 99.39% | 99.12% | 98.99% | 95.53% | 72.06% | Capsnet | 99.17% | 98.77% | 98.73% | 98.29% | 79.18% |
| EMcaps | 98.84% | 98.79% | 98.55% | 94.03% | 70.03% | EMcaps | 98.83% | 98.38% | 98.42% | 97.86% | 77.47% |
| G-Caps | 97.44% | 96.31% | 96.01% | 93.18% | **81.70%** | G-Caps | 97.67% | 96.53% | 96.33% | 95.52% | 83.76% |
| DeepCaps | 99.54% | 99.49% | 99.42% | 97.27% | 73.61% | DeepCaps | 99.44% | 99.16% | 99.03% | 98.64% | 77.54% |
| GCNN | 99.52% | 99.38% | 99.37% | 97.02% | 74.98% | GCNN | 89.34% | 89.16% | 89.13% | 88.86% | 75.53% |
| SOVNET | **99.70%** | **99.65%** | **99.63%** | **98.56%** | 79.59% | SOVNET | **99.68%** | **99.60%** | **99.59%** | **99.5%** | **87.76%** |

| Results on Training on MNIST Transformed by (2,180°) | | | | |
|---|---|---|---|---|
| Method | (0,0°) | (2,30°) | (2,60°) | (2,90°) | (2,180°) |
| Capsnet | 97.52% | 96.65% | 96.64% | 96.50% | 96.09% |
| EMcaps | 95.89% | 95.22% | 95.42% | 95.42% | 95.09% |
| G-Caps | 95.24% | 93.67% | 93.83% | 93.79% | 93.76% |
| DeepCaps | 98.17% | 97.84% | 97.89% | **98.11%** | 98.01% |
| GCNN | 87.8% | 87.51% | 87.47% | 87.41% | 87.45% |
| SOVNET | **98.34%** | **98.10%** | **98.11%** | 98.08% | **98.06%** |

On the CIFAR-10 experiments, DeepCaps performs significantly better than SOVNET on the untransformed case - generalising to test-time transformations better. However, SOVNET learns from train-time transformations better than DeepCaps - outperforming it in a large majority of the other cases. We hypothesize that the first observation is due to the increased (almost double) number of parameters of DeepCaps that allows it to learn features that generalise better to transformations. Further, as p4-convolutions (the prediction-mechanisms used) are equivariant only to rotations in multiples of 90°, its performance is significantly lower for test-time transformations of 30°and 60°for the untransformed case. However, the equivariance of SOVNET allows it to learn better from train-time geometric transforms than DeepCaps, explaining the second observation.

The second case is that GCaps outperforms SOVNET on generalising to extreme transformations on (mainly) MNIST, and once on FashionMNIST, under mild train-time conditions. However, it is unable to sustain this under more extreme train-time perturbations. We infer that this is caused largely by the explicit geometric parameterisation of capsules in G-Caps. While under mild-to-moderate train-time conditions, and on simple datasets, this approach could yield better results, this parameterisation, especially with very simple prediction-mechanisms, can prove detrimental. Thus, the convolutional nature of the prediction-mechanisms, which can capture more complex features, and also the greater depth of SOVNET allows it to learn better from more complex training scenarios. This makes the case for deeper models with more expressive and equivariant prediction-mechanisms.

A related point of interest is that G-Caps performs very poorly on the CIFAR-10 dataset - achieving the least accuracy on most cases on this dataset - despite provable guarantees on equivariance. We argue that this is significantly due to the nature of the capsules of this model itself. In GCaps, each capsule is explicitly modelled as an element of a Lie group. Thus, capsules capture exclusively geometric information, and use only this information for routing. In contrast, other capsule models have no such parameterisation. In the case of CIFAR-10, where non-geometric features such as

Table 3: Experiments on FashionMNIST.

| Results on Training on Untransformed FashionMNIST | | | | | | Results on Training on FashionMNIST Transformed by (2,30°) | | | | |
|---|---|---|---|---|---|---|---|---|---|---|
| Method | (0,0°) | (2,30°) | (2,60°) | (2,90°) | (2,180°) | Method | (0,0°) | (2,30°) | (2,60°) | (2,90°) | (2,180°) |
| Capsnet | 91.23% | 57.15% | 37.98% | 28.33% | 22.38% | Capsnet | 91.22% | 89.57% | 69.58% | 50.17% | 35.16% |
| EMcaps | 90.05% | 59.75% | 40.26% | 30.17% | 23.82% | EMcaps | 90.17% | 89.47% | 68.39% | 49.23% | 37.02% |
| G-Caps | 86.56% | 50.05% | 35.05% | 29.93% | 27.10% | G-Caps | 83.28% | 80.12% | 64.86% | 53.71% | **52.54%** |
| DeepCaps | 93.27% | 57.85% | 37.06% | 27.63% | 21.86% | DeepCaps | 93.71% | 93.40% | 75.32% | 53.35% | 36.30% |
| GCNN | 84.63%56.37% | 31% | 0.2862% | 21.58% | | GCNN | 92.25% | 90.95% | 72.17% | 51.93% | 37.12% |
| SOVNET | **94.72%** | **61.58%** | **41.01%** | **34.07%** | **27.63%** | SOVNET | **94.99%** | **94.36%** | **77.19%** | **58.59%** | **43.84%** |

| Results on Training on FashionMNIST Transformed by (2,60°) | | | | | | Results on Training on FashionMNIST Transformed by (2,90°) | | | | |
|---|---|---|---|---|---|---|---|---|---|---|
| Method | (0,0°) | (2,30°) | (2,60°) | (2,90°) | (2,180°) | Method | (0,0°) | (2,30°) | (2,60°) | (2,90°) | (2,180°) |
| Capsnet | 89.98% | 88.55% | 88.15% | 72.81% | 46.89% | Capsnet | 88.78% | 87.18% | 87.13% | 86.19% | 59.59% |
| EMcaps | 88.24% | 87.30% | 87.04% | 71.72% | 48.14% | EMcaps | 86.43% | 85.85% | 85.82% | 85.63% | 61.15% |
| G-Caps | 82.04% | 80.12% | 78.94% | 68.05% | 59.25% | G-Caps | 80.71% | 79.55% | 79.17% | 79.21% | 72.11% |
| DeepCaps | 93.36% | 93.06% | 92.84% | 80.76% | 49.90% | DeepCaps | 93.07% | 92.93% | 92.75% | 92.51% | 62.50% |
| GCNN | 90.78% | 89.82% | 89.67% | 76.69% | 49.97& | GCNN | 90.31% | 89.46% | 89.42% | 89.22% | 64.44% |
| SOVNET | **94.49%** | **94.08%** | **94.20%** | **90.23%** | **73.48%** | SOVNET | **94.41%** | **94.03%** | **93.93%** | **93.98%** | **91.42%** |

| Results on Training on FashionMNIST Transformed by (2,180°) | | | | | |
|---|---|---|---|---|---|
| Method | (0,0°) | (2,30°) | (2,60°) | (2,90°) | (2,180°) |
| Capsnet | 86.90% | 84.94% | 84.93% | 84.75% | 84.72% |
| EMcaps | 82.99% | 82.67% | 82.18% | 82.32% | 82.18% |
| G-Caps | 80.65% | 79.66% | 79.46% | 79.47% | 79.37% |
| DeepCaps | 92.07% | 91.71% | 91.70% | 91.76% | 91.66% |
| GCNN | 89.7% | 88.65% | 88.61% | 88.62% | 88.6% |
| SOVNET | **94.11%** | **93.77%** | **93.56%** | **93.57%** | **93.60%** |

Table 4: Experiments on Cifar-10.

| Results on Training on Untransformed Cifar-10 | | | | | | Results on Training on Cifar-10 Transformed by (2,30°) | | | | |
|---|---|---|---|---|---|---|---|---|---|---|
| Method | (0,0°) | (2,30°) | (2,60°) | (2,90°) | (2,180°) | Method | (0,0°) | (2,30°) | (2,60°) | (2,90°) | (2,180°) |
| Capsnet | 68.28% | 55.57% | 43.55% | 37.48% | 30.89% | Capsnet | 73.45 | 69.87% | 61.17% | 52.29% | 42.58% |
| EMcaps | 62.85% | 49.28% | 41.37% | 34.73% | 29.90% | EMcaps | 70.24% | 66.63% | 59.10% | 50.93% | 42.26% |
| G-Caps | 49.54% | 38.45% | 31.89% | 30.88% | 27.70% | G-Caps | 49.50% | 48.88% | 45.78% | 42.93% | 38.74% |
| DeepCaps | 76.76% | **67.97%** | **53.56%** | **45.22%** | 35.67% | DeepCaps | 84.24% | 82.54% | 74.63% | 63.54% | 48.63% |
| GCNN | 80.26% | 60.94% | 47.93% | 40.85% | 34.02% | GCNN | 82.13% | 78.94% | 70.51% | 59.80% | 47.81% |
| SOVNET | **88.34%** | 47.57% | 42.24% | 43.75% | **43.52%** | SOVNET | **86.58%** | **85.35%** | **82.51%** | **79.14%** | **69.64%** |

| Results on Training on Cifar-10 Transformed by (2,60°) | | | | | | Results on Training on Cifar-10 Transformed by (2,90°) | | | | |
|---|---|---|---|---|---|---|---|---|---|---|
| Method | (0,0°) | (2,30°) | (2,60°) | (2,90°) | (2,180°) | Method | (0,0°) | (2,30°) | (2,60°) | (2,90°) | (2,180°) |
| Capsnet | 70.26% | 67.69% | 66.62% | 60.04% | 47.99% | Capsnet | 67.81% | 65.64% | 65.46% | 64.35% | 52.79% |
| EMcaps | 66.53% | 65.09% | 63.21% | 58.04% | 47.61% | EMcaps | 64.33% | 63.00% | 62.70% | 61.42% | 52.08% |
| G-Caps | 49.63% | 50.31% | 48.84% | 47.43% | 43.11% | G-Caps | 49.98% | 51.24% | 50.63% | 49.95% | 46.59% |
| DeepCaps | 83.92% | 83.63% | **82.79%** | 78.09% | 60.02% | DeepCaps | 82.91% | **82.78%** | **82.66%** | 82.62% | 68.34% |
| GCNN | 82.13% | 78.94% | 70.51% | 59.80% | 47.81% | GCNN | 77.56% | 75.79% | 75.57% | 75.14% | 64.23% |
| SOVNET | **83.86%** | **83.63%** | 83.57% | **83.06%** | **80.89%** | SOVNET | **83.33%** | 82.76% | 82.58% | **82.79%** | **82.22%** |

| Results on Training on Cifar-10 Transformed by (2,180°) | | | | | |
|---|---|---|---|---|---|
| Method | (0,0°) | (2,30°) | (2,60°) | (2,90°) | (2,180°) |
| Capsnet | 61.08% | 59.53% | 60.04% | 59.85% | 59.90% |
| EMcaps | 57.57% | 55.89% | 56.85% | 56.35% | 55.20% |
| G-Caps | 39.09% | 41.03% | 41.43% | 41.25% | 41.08% |
| DeepCaps | 81.12% | 80.81% | 80.64% | 81.05% | 80.92% |
| GCNN | 73.61% | 72.50% | 72.12% | 72.24% | 72.25% |
| SOVNET | **82.50%** | **81.80%** | **81.78%** | **81.95%** | **81.82%** |

texture are important, we see that purely spatio-geometric based routing is not effective. This observation allows us to make a more general hypothesis that could deal with the fundamentals of capsule networks. We propose a trade-off in capsule networks, based on the notion of equivariance. To appreciate this, some background is necessary on both equivariance and capsule networks.

As the body of literature concerning equivariance is quite vast, we only mention a relevant selection of papers. Equivariance can be seen as a desirable, if not fundamental, inductive bias for neural networks used in computer vision. Indeed, the fact that AlexNet (Krizhevsky et al., 2012) automatically learns representation that are equivariant to flips, rotation and scaling shows the importance of equivariance as well as its natural necessity (Lenc & Vedaldi, 2015). Thus, a neural network model that can formally guarantee this property is essential. An early work in this regard is the group-equivariant convolution proposed in (Cohen & Welling, 2016). There, the authors proposed a generalisation of the 2-D spatial convolution operation to act on a general group of symmetry transforms - increasing the parameter-sharing and, thereby, improving performance. Since then,

Table 5: Results on augmented SVHN and KMNIST.

| Method | SVHN | KMNIST |
|---|---|---|
| Sabour et al. (2017) | 95.7% | - |
| Deliège et al. (2018) | 94.50% | - |
| Rajasegaran et al. (2019) | **97.16%** | 89.18% |
| Phaye et al. (2018) | 96.90% | - |
| Clanuwat et al. (2018) | - | 98.83% |
| Tissera et al. (2019) | 96.8% | **99.05%** |
| SOVNET | **97.03%** | **99.03%** |

several other models exhibiting equivariance to certain groups of transformations have been proposed, for example (Cohen et al., 2018b), where a spherical correlation operator that exhibits rotation-equivariance was introduced; (Carlos Esteves & Daniilidis, 2017), where a network equivariant to rotation and scale, but invariant to translations was presented, and Worrall & Brostow (2018), where a model equivariant to translations and 3D right-angled rotations was developed. A general theory of equivariant CNNs was developed in (Cohen et al., 2018a). In their paper, they show that convolutions with equivariant kernels are the most general class of equivariant maps between feature spaces.

A fundamental issue with group-equivariant convolutional networks is the fact that the grid the convolution works with increases exponentially with the type of the transformations considered. This was pointed out in (Sabour et al., 2017); capsules were proposed as an efficient alternative. In a general capsule network model, each capsule is supposed to represent the pose-coordinates of an object-component. Thus, to increase the scope of equivariance, only a linear increase in the dimension of each capsule is necessary. This was however not formalised in most capsule architectures, which focused on other aspects such as routing (Hinton et al., 2018), (Bahadori, 2018), (Wang & Liu, 2018); general architecture (Rajasegaran et al., 2019), (Deliège et al., 2018), (Rawlinson et al., 2018), Jeong et al. (2019), (Phaye et al., 2018), Rosario et al. (2019); or application Afshar et al. (2018).

It was only in group-equivariant capsules (Lenssen et al., 2018) that this idea of efficient equivariance was formalised. Indeed, in that paper, equivariance changed from preserving the action of a group on a vector space to preserving the group-transformation on an element. While such models scale well to larger transformation groups in the sense of preserving equivariance guarantees, we argue that they cannot efficiently handle compositionalities that involve more than spatial geometry. The direct use of capsules as geometric pose-coordinates could lead to exponential representational inefficiencies in the number of capsules. This is the tradeoff we referred to. We do not attempt a formalisation of this, and instead make the observation given next. While SOVNET (using GCNNs) lacks in transformational efficiency, the use of convolutions allows it to capture non-geometric structures well. Further, SOVNET still retains the advantage of learning compositional structures better than CNN models due to the use of routing, placing it in a favourable position between two extremes.

## 5 CONCLUSION

We presented a scalable, equivariant model for capsule networks that uses group-equivariant convolutions and degree-centrality routing. We proved that the model preserves detected compositionalities under transformations. We presented the results of experiments on affine variations of various classification datasets, and showed that our model performs better than several capsule network baselines. A second set of experiments showed that our model performs comparably to convolutional baselines on two other datasets. We also discussed a possible tradeoff between efficiency in the transformational sense and efficiency in the representation of non-geometric compositional relations. As future work, we aim at understanding the role of the routing algorithm in the optimality of the capsule-decomposition graph, and various other properties of interest based on it. We also note that SOVNET allows other equivariant prediction mechanisms - each of which could result in a wider application of SOVNET to different domains.

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

## APPENDIX

## ACKNOWLEDGEMENTS

We thank Vineeth Balasubramanian for the discussions we have had, and his useful comments. We also thank Darshan Gera, Rishi Rao and Ankit Anand for their suggestions and comments.

## A    DEFINITIONS

We present formal definitions for the following concepts.

## A.1   GROUP

A tuple $(G, \circ)$, where $G$ is a non-empty set and $\circ$ defines a binary operation on $G$, is said to form a group if the following properties are satisfied:

**Closure:** $\forall\, g_1, g_2 \in G,\, g_1 \circ g_2 \in G$.

**Associativity:** $\forall\, g_1, g_2, g_3 \in G,\, (g_1 \circ g_2) \circ g_3 = g_1 \circ (g_2 \circ g_3)$.

**Existence of the identity element:** $\exists\, e \in G \ni \forall\, g \in G,\, e \circ g = g \circ e = g$.

**Existence of an inverse:** $\forall\, g \in G,\, \exists\, g^{-1} \in G \ni g \circ g^{-1} = g^{-1} \circ g = e$.

## A.2   GROUP ACTION AND GROUP REPRESENTATION

Given a group $(G, \circ)$ and a vector space $V$, a group action is a function $f : G \times V \to V$ satisfying the following properties.

**1:** $\forall\, a \in V,\, f(e, a) = a$.

**2:** $\forall\, g, h \in G$ and $\forall\, a \in V,\, f(h, f(g, a)) = f(h \circ g, a)$.

A group representation is a group action by invertible linear maps. More formally, a group representation of a group $(G, \circ)$ with respect to a vector space $V$ is a homomorphism from $G$ to $GL(V)$ - the set of linear, invertible maps from $V$ to $V$.

## A.3   POOL

Consider a one-layer GCNN-convolutional prediction network $\Psi_j^{l+1}$ for a SOVNET layer $l + 1$, and for the $d^{l+1}$- dimensional $j^{th}$ capsule-type. Intuitively, $Pool_j^{l+1}(g)$ is defined by the extent of the support of the g-transformed filter $\Psi_j^{l+1}$. More formally,

$$Pool_j^{l+1}(g) = \{h \in G : \Psi_j^{l+1}(g^{-1} \circ h) \neq 0\}. \tag{1}$$

For a general $L$-layer GCNN prediction- network, $Pool_j^{l+1}(g)$ is defined by recursively applying the above definition through all the layers of the prediction network.

## A.4   2-NORM

The 2-norm of a vector $x = (x_0, ..., x_{n-1})^T \in \mathbb{R}^n$, and denoted by $\|x\|_2$, is defined as $\|x\|_2 = (\sum_{i=0}^{n-1} x_i^2)^{\frac{1}{2}}$.

## B   PROOFS OF THEOREMS

We present proofs for the theorems mentioned in the main body.

**Theorem B.1.** *The SOVNET layer defined in Algorithm 2, and denoted by the operator $\otimes$ as given above, satisfies $([L_g F^l] \otimes \Psi_j^{l+1}) = (L_g[F^l \otimes \Psi_j^{l+1}])$, where g belongs to the underlying group of the equivariant convolution.*

*Proof.* For the theorem to be true, we must show that each step of Algorithm 2 is equivariant. We do this step-wise.

The predictions $S_{ij}^{l+1}$ made in the first step are group-equivariant. This follows from the fact that $S_{ijp}^{l+1}(g) = (f_i^l \star \Psi_j^{l+1,p})(g)$, and that $([L_h f_i^l] \star \Psi_j^{l+1,p}) = L_h(f_i^l \star \Psi_j^{l+1,p})$ - proved in (Cohen & Welling, 2016).

We now show that the $DegreeScore$ procedure is equivariant. We see that $Degree_i^j(g)$ $= \sum_{k=0}^{N_l-1}(\frac{S_{ij}^{l+1}(g).S_{kj}^{l+1}(g)}{\|S_{ij}^{l+1}(g)\|_2.\|S_{kj}^{l+1}(g)\|_2})$; $0 \leq i \leq N_l - 1$. Each $S_{ij}^{l+1}(g).S_{kj}^{l+1}(g) = \sum_{p=0}^{d^{l+1}-1}(f_i^l \star \Psi_j^{l+1,p})(g)(f_k^l \star \Psi_j^{l+1,p})(g)$. From the equivariance of $\star$, $\sum_{p=0}^{d^{l+1}-1}([L_h f_i^l] \star \Psi_j^{l+1,p})(g)([L_h f_k^l] \star \Psi_j^{l+1,p})(g) = \sum_{p=0}^{d^{l+1}-1} L_h(f_i^l \star \Psi_j^{l+1,p})(g)L_h(f_k^l \star \Psi_j^{l+1,p})(g) = \sum_{p=0}^{d^{l+1}-1}(f_i^l \star \Psi_j^{l+1,p})(h^{-1} \circ g)(f_k^l \star \Psi_j^{l+1,p})(h^{-1} \circ g) = [\sum_{p=0}^{d^{l+1}-1}(f_i^l \star \Psi_j^{l+1,p})(f_k^l \star \Psi_j^{l+1,p})](h^{-1} \circ g) = L_h[\sum_{p=0}^{d^{l+1}-1}(f_i^l \star \Psi_j^{l+1,p})(f_k^l \star \Psi_j^{l+1,p})](g)$.

Moreover, the 2-norm of an equivariant map is also equivariant - from the equivariance of the post-composition of non-linearities over equivariant maps (Cohen & Welling, 2016). Also, the division of two (non-zero) equivariant maps is also equivariant. Thus, obtaining the degree-scores is equivariant. Again, the softmax function preserves the equivariance as it is a point-wise non-linearity.

The proof is concluded by pointing out that the product and sum of equivariant maps is also equivariant. □

**Theorem B.2.** *Consider an L-layer SOVNET whose activations are routed according to a procedure belonging to the family given by Algorithm 1. Further, assume that this routing procedure is equivariant with respect to the group G. Then, given an input x and $\forall g \in G$, $G(x)$ and $G([L_g x])$ are isomorphic.*

*Proof.* Consider a fixed $L$-layer SOVNET that is equivariant to transformations from a group $G$, and an input $x : G \to \mathbb{R}^c$. Let $\mathbb{G}(x)$ be the capsule-decomposition graph corresponding to $x$. Then $\mathbb{G}(L_h x)$ denotes the the capsule-decomposition graph of the transformed input $L_h x$.

We show that the map $\tilde{f}_i^l(g) \to \tilde{f}_i^l(h^{-1} \circ g)$ is an isomorphism from $\mathbb{G}(x)$ to $\mathbb{G}(L_h x)$. First, we note that $\tilde{f}_i^l(g) \to \tilde{f}_i^l(h^{-1} \circ g)$ is a bijection from $V(x)$ to $V(L_h x)$. This is from the definition of the vertex set of a capsule-decomposition graph and the fact that the map $g \to h^{-1} \circ g$ is a bijection.

We now show that $(\tilde{f}_i^l(g_1), \tilde{f}_j^{l+1}(g_2), c_{ij}(g_2)) \in E(x)$ if and only if $(\tilde{f}_i^{l+1}(h^{-1} \circ g_1), \tilde{f}_j^{l+1}(h^{-1} \circ g_2), c_{ij}(h^{-1} \circ g_2)) \in E(L_h x)$.

First, let us assume $(\tilde{f}_i^l(g_1), \tilde{f}_j^{l+1}(g_2), c_{ij}(g_2)) \in E(x)$. Thus, $\tilde{f}_i^l(g_1)$ is routed to $\tilde{f}_j^l(g_2)$ with routing-coefficient $c_{ij}(g_2)$. However, due to the assumed equivariance of the model, $\tilde{f}_i^{l+1}(h^{-1} \circ g_1)$ is routed to $\tilde{f}_j^{l+1}(h^{-1} \circ g_2)$ with routing-coefficient $c_{ij}(h^{-1} \circ g_2)$. This, of course, implies $(\tilde{f}_i^{l+1}(h^{-1} \circ g_1), \tilde{f}_j^{l+1}(h^{-1} \circ g_2), c_{ij}(h^{-1} \circ g_2)) \in E(L_h x)$.

The converse of this result is proved in the same way by considering $E(L_h x)$, noting that $E(L_{h^{-1}} L_h x) = E(x)$, and applying the above result to $E(L_h x)$ and $E(L_{h^{-1}} L_h x)$. □

## C  FURTHER EXPERIMENTS

### C.1  EMPIRICAL VALIDATION OF CAPSULE DECOMPOSITION ISOMORPHISM

We performed two experiments to verify that the capsule decomposition-graphs of the transformed and untransformed images are isomorphic.

For the first of these, we trained a p4-convolution based SOVNET architecture on untransformed images of MNIST and FashionMNIST. We then considered four variations of the two test-datasets - untransformed, and three versions rotated exactly by multiples of 90 degrees: 90, 180, and 270. Our experiment verifies that the mapping defined in the proof of Theorem 2.2 is indeed an isomorphism.

To this end, we considered the capsule-activations as well as the degree-scores, obtained across all the capsule-layers, for each image of all the variations of the test split of the corresponding dataset. We then mapped the activations and the degree-scores for the untransformed images by the aforesaid mapping for each of the transformations. This corresponds to 'rotating' the activations and degree-scores by each transformation. We then computed the squared error of these with each of the activations and degree-scores obtained from the correspondingly transformed image, respectively. A successful verification would result in zero error (up to machine precision). The results in Table 6 show that this happens.

The second of our experiments is an empirical verification that the test-accuracies remain unchanged under transformations for which SOVNET exhibits equivariance. We use the same trained architecture as above, and verify that the accuracy remains unchanged under exact transformations of the images. The results are presented in Table 7. The accuracies presented below are only for the purpose of veryfying the isomorphism of the of the graph.

Table 6: Empirical validation of isomorphism: mean squared error of capsule activations and degree-scores.

| MNIST | | |
|---|---|---|
| Rotation | Mean squared-error for capsules | Mean squared-error for degree-scores |
| 90 | 6.1900e-15 | 3.3087e-15 |
| 180 | 6.2821e-15 | 3.3606e-15 |
| 270 | 6.1911e-15 | 3.3138e-15 |
| FashionMNIST | | |
| Rotation | Mean squared-error for capsules | Mean squared-error for degree-scores |
| 90 | 2.5678e-13 | 1.9576e-13 |
| 180 | 2.6306e-13 | 1.9981e-13 |
| 270 | 2.5869e-13 | 1.9662e-13 |

| CIFAR10 | | |
|---|---|---|
| Rotation | Mean squared-error for capsules | Mean squared-error for degree-scores |
| 90 | 6.4583e-13 | 2.1735e-13 |
| 180 | 6.3866e-13 | 2.1635e-13 |
| 270 | 6.4624e-13 | 2.1774e-13 |

Table 7: Empirical validation of isomorphism: accuracies under transformation.

| Accuracies on MNIST | | | |
|---|---|---|---|
| 0° | 90° | 180° | 270° |
| 99.52% | 99.52% | 99.52% | 99.52% |

| Accuracies on FashionMNIST | | | |
|---|---|---|---|
| 0° | 90° | 180° | 270° |
| 92.23% | 92.23% | 92.23% | 92.23% |

| Accuracies on CIFAR10 | | | |
|---|---|---|---|
| 0° | 90° | 180° | 270° |
| 77.19% | 77.19% | 77.19% | 77.19% |

## C.2    RESULTS ON TESTING ON UNSEEN TRANSFORMS: AFFNIST

We trained a SOVNET architecture on MNIST images that are padded to size 40x40 - the size of AFFNIST images. We augment these images by translation, as is the standard approach. Note that the changed size of the images necessitates a different architecture. The result of this experiment is given in Table 8. We see that our SOVNET architecture obtains the highest accuracy when compared to other recent capsule network models.

Table 8: Accuracy on AFFNIST.

| Method | Accuracy |
|---|---|
| (Sabour et al., 2017) | 79.0% |
| (Hinton et al., 2018) | 93.1% |
| (Lenssen et al., 2018) | 89.10% |
| (Jeong et al., 2019) | 87.8% |
| (Choi et al., 2019) | 91.6% |
| **SOVNET** | **97.01**% |

We also trained the above SOVNET architecture on MNIST with translations in the range of [-6,6] pixels and rotations from [-30 ,30] degrees. While this increases the extent of train-time augmentation, there are several test-time transformations that are unseen. With this scheme, we achieve state-of-the-art accuracy of 99.20%. This improves over the best, to our knowledge, accuracy of 98.3% obtained by (Tai et al., 2019).

## C.3 CAPSNET WITH SHARED PARAMETERS

We considered an implementation of the CAPSNET model (Sabour et al., 2017). Unlike (Sabour et al., 2017), that uses one prediction-network per connection between capsules, this model uses one prediction-network per class-capsule. The result of this model on augmented versions of MNIST and FashionMNIST are presented in Table 9, with corresponding accuracies of capsnet.

Table 9: Results on MNIST and FashionMNIST.

| Method | MNIST | FashionMNIST |
|---|---|---|
| Capsnet | 99.75%(Sabour et al., 2017) | 93.62%(Rajasegaran et al., 2019) |
| Shared Capsnet | 99.47% | 91.57% |

## C.4 RESULTS OF SOVNET ON CIFAR-100

We have trained a SOVNET architecture on CIFAR100. Our model has achieved an accuracy of 71.55%, an almost 4 percentage improvement over a recent capsule network model - STARCAPS (Karim Ahmed, 2019) which achieved 67.66%.

## C.5 COMPARISON AGAINST RESNET ARCHITECTURES

In order to compare SOVNET with more sophisticated CNN models, we performed a limited set of experiments on MNIST and FashionMNIST. We trained ResNet18 and ResNet34 on the train split of MNIST and FashionMNIST transformed by random translations of up to $\pm$ 2 pixels, and random rotations of up to $\pm$ 180°. The models were tested on various transformed versions of the test-splits. The results of these experiments are given in Table 10. As can be seen in the table, SOVNET compares with the two much deeper CNN models. More testing on more complex datasets, as well as deeper SOVNET models must be done, however, to obtain a better understanding of the relative performance of these two kinds of models.

## D DISCUSSION ON GENERAL WEIGHTED-SUMMATION ROUTING

Consider Algorithm 1, which is given below for convenience. The role of the $GetWeights$ and $Agreement$ procedures is to evaluate the relative importances of predictions for a deeper capsule,

Table 10: Results on ResNet18 and ResNet34.

| Results on Training on MNIST Transformed by (2,180°) | | | | | Results on Training on FashionMNIST Transformed by (2,180°) | | | | |
|---|---|---|---|---|---|---|---|---|---|
| Method | (0,0°) | (2,30°) | (2,60°) | (2,90°) | (2,180°) | Method | (0,0°) | (2,30°) | (2,60°) | (2,90°) | (2,180°) |
| ResNet18 | 98.60% | 98.30% | 98.21% | 98.15% | 98.02% | ResNet18 | 94.21% | 93.55% | 93.24% | 93.30% | 93.45% |
| ResNet34 | 98.53% | 98.26% | 98.21% | 98.12% | 98.01% | ResNet34 | 94.38% | 93.75% | 93.78% | 93.78% | 93.73% |
| SOVNET | 98.34% | 98.10% | 98.11% | 98.08% | 98.06% | SOVNET | 94.11% | 93.77% | 93.56% | 93.57% | 93.60% |

and the extent of consensus among them, respectively. The second of these is interpreted as a measure of the activation of the corresponding deeper capsule.

A formalisation of these concepts to a general framework for even summation-based routing so as to cover all possible notions of relative importance, and consensus is not within the scope of this paper. Indeed, to the best of our knowledge, such a formalisation has not been successfully completed.

Thus, instead of a formal description of a general routing procedure, we provide examples to better understand the role of these two functions. We first explain $GetWeights$, and then $Agreement$.

---

**Algorithm** A general weighted-summation routing algorithm for SOVNET.

---

**Input**: $\{(f_i^l, a_i^l)|i \in \{0, ..., N_l - 1\}, f_i^l : G \to \mathbb{R}^{d^l}, a_i^l : G \to [0,1]\}$
**Output**: $\{(f_j^{l+1}, a_j^{l+1})|j \in \{0, ..., N_{l+1} - 1\}, f_j^{l+1} : G \to \mathbb{R}^{d^{l+1}}, a_j^{l+1} : G \to [0,1]\}$
**Trainable Functions**: $(\Psi_j^{l+1}, \cdot)$ - projection networks that use operator $\cdot$

$\quad S_{ij}^{l+1}(g) = ((f_i^l, a_i^l) \cdot \Psi_j^{l+1})(g) \; \forall \, i, j, \forall g \in G$

$\quad (c_{0j}^{l+1}(g), ..., c_{N_l-1j}^{l+1}(g)) = GetWeights(S_{0j}^{l+1}(g), ..., S_{N_l-1j}^{l+1}(g)) \; \forall \, j, \forall g \in G$

$\quad f_j^{l+1}(g) = \sum_{i=1}^{N_l-1} c_{i,j}^{l+1}(g) S_{ij}^{l+1}(g) \; \forall \, j, \forall g \in G$

$\quad a_j^{l+1}(g) = Agreement(f_j^{l+1}(g), S_{0j}^{l+1}(g), ..., S_{N_I-1j}^{l+1}(g)) \; \forall \, j$

---

The first example of $GetWeights$ we provide is from the proposed degree-centrality based routing. The algorithm is given below, again. In this case, $GetWeights$ is instantiated by the $DegreeScore$ procedure, which assigns weights to predictions based on their normalised degree centrality scores. Thus, a prediction that agrees with a significant number of its peers obtains a higher importance than one that does not. This scheme follows the principle of routing-by-agreement, that aims to activate a deeper capsule only when its predicting shallower, component-capsules are in an acceptable spatial configuration (Hinton et al., 2011).

The above form for the summation-based routing procedure generalises for several existing routing algorithms. As an example, we present the dynamic routing algorithm of (Sabour et al., 2017). This differs with our proposed algorithm in that it is a "attention-based", rather than "agreement-based" routing algorithm. That is, the relative importance of a prediction with respect to a fixed deeper capsule is not a direct measure of the extent of its consensus with its peers, but rather a measure of the relative attention it offers to the deeper capsule.

Thus, the weight associated with a prediction for a fixed deeper capsule by a fixed shallower capsule depends on other deeper capsules. In order to accomodate such methods into a general procedure, we modify our formalism by having $GetWeights$ take all the predictions as parameters, and return all the routing weights. This modified general procedure is given in Algorithm 5.

Consider the dynamic routing algorithm of (Sabour et al., 2017), given in Algorithm 6 - modified to our notation and also the use of group-equivariant convolutions. The procedure $DynamicRouting$ is the instantiation for $GetWeights$. Note that the weights $c_{ij}(g)$ depend on the routing weights for the deeper capsules.

Due to the formulation of capsules in our paper, as in (Sabour et al., 2017), we use the 2-norm of a capsule to denote its activation. Thus, our degree-centrality based procedure, and also dynamic routing, do not use a separate value for this. However, examples of algorithms that use a separate

**Algorithm** The degree-centrality based routing algorithm for SOVNET.

---

**Input**: $\{f_i^l | i \in \{0, ..., N_l - 1\}, f_i^l : G \to \mathbb{R}^{d^l}\}$

**Output**: $\{f_j^{l+1} | j \in \{0, ..., N_{l+1} - 1\}, f_j^{l+1} : G \to \mathbb{R}^{d^{l+1}}\}$

**Trainable Functions**: $(\Psi_j^{l+1}, \star), 0 \le j \le N_{l+1} - 1$, - a set of $d^{l+1}$ group-equivariant convolutional filters (per capsule-type) that use the group-equivariant correlation operator $\star$

$S_{ijp}^{l+1}(g) = (f_i^l \star \Psi_j^{l+1,p})(g) = \sum_{h \in G} \sum_{k=0}^{d^l-1} f_{ik}^l(h) \Psi_k^{l+1,p}(g^{-1} \circ h); p \in \{0, ..., d^{l+1} - 1\}$

$(c_{0j}^{l+1}(g), ..., c_{N_l-1j}^{l+1}(g)) = DegreeScore(S_{0j}^{l+1}(g), ..., S_{N_l-1j}^{l+1}(g)); \forall 0 \le j \le N_{l+1} - 1, \forall g$

$f_j^{l+1}(g) = \sum_{i=0}^{N_l-1} c_{ij}^{l+1}(g) S_{ij}^{l+1}(g) \forall 0 \le j \le N_{l+1} - 1, \forall g \in G$

$f_j^{l+1}(g) = Squash(f_j^{l+1}(g)) = \frac{\|f_j^{l+1}(g)\|_2}{1 + \|f_j^{l+1}(g)\|_2^2} f_j^{l+1}(g); \forall 0 \le j \le N_l - 1, \forall g \in G$

    **procedure** DEGREESCORE($S_{0j}^{l+1}(g), ..., S_{N_l-1j}^{l+1}(g)$)

$A_{ik}^j(g) = \frac{S_{ij}^{l+1}(g).S_{kj}^{l+1}(g)}{\|S_{ij}^{l+1}(g)\|_2 . \|S_{kj}^{l+1}(g)\|_2}; 0 \le i, k \le N_l - 1$

$Degree_i^j(g) = \sum_{k=0}^{N_l-1}(A_{ik}^j(g)); 0 \le i \le N_l - 1$

$c_{ij}(g) = \frac{\exp(Degree_i^j(g))}{\sum_{k=0}^{N_l-1} \exp(Degree_k^j(g))}; 0 \le i \le N_l - 1$

    **return** $c_{ij}(g) \forall 0 \le i \le N_l - 1$

---

**Algorithm 5** A general weighted-summation routing algorithm for SOVNET.

---

**Input**: $\{(f_i^l, a_i^l) | i \in \{0, ..., N_l - 1\}, f_i^l : G \to \mathbb{R}^{d^l}, a_i^l : G \to [0, 1]\}$

**Output**: $\{(f_j^{l+1}, a_j^{l+1}) | j \in \{0, ..., N_{l+1} - 1\}, f_j^{l+1} : G \to \mathbb{R}^{d^{l+1}}, a_j^{l+1} : G \to [0, 1]\}$

**Trainable Functions**: $(\Psi_j^{l+1}, \cdot)$ - projection networks that use operator $\cdot$

$S_{ij}^{l+1}(g) = ((f_i^l, a_i^l) \cdot \Psi_j^{l+1})(g) \forall i, j, \forall g \in G$

$(c_{00}^{l+1}(g), ..., c_{N_l-1N_{l+1}-1}^{l+1}(g)) = GetWeights(S_{00}^{l+1}(g), ..., S_{N_l-1N_{l+1}-1}^{l+1}(g)) \forall g \in G$

$f_j^{l+1}(g) = \sum_{i=1}^{N_l-1} c_{i,j}^{l+1}(g) S_{ij}^{l+1}(g) \forall j, \forall g \in G$

$a_j^{l+1}(g) = Agreement(f_j^{l+1}(g), S_{0j}^{l+1}(g), ..., S_{N_I-1j}^{l+1}(g)) \forall j$

---

**Algorithm 6** The dynamic routing algorithm

---

**Input**: $\{f_i^l | i \in \{0, ..., N_l - 1\}, f_i^l : G \to \mathbb{R}^{d^l}\}$

**Output**: $\{f_j^{l+1} | j \in \{0, ..., N_{l+1} - 1\}, f_j^{l+1} : G \to \mathbb{R}^{d^{l+1}}\}$

**Trainable Functions**: $(\Psi_j^{l+1}, \star), 0 \le j \le N_{l+1} - 1$, a set of $d^{l+1}$ group-equivariant convolutional filters (per capsule-type) that use the group-equivariant correlation operator $\star$

$S_{ijp}^{l+1}(g) = (f_i^l \star \Psi_j^{l+1,p})(g) = \sum_{h \in G} \sum_{k=1}^{d^l-1} f_{ik}^l(h) \Psi_k^{l+1,p}(g^{-1} \circ h); p \in \{0, ..., d^{l+1} - 1\}$

$(c_{00}^{l+1}(g), ..., c_{N_l-1N_{l+1}-1}^{l+1}(g)) = DynamicRouting(S_{00}^{l+1}(g), ..., S_{N_l-1N_{l+1}-1}^{l+1}(g)); \forall g$

$f_j^{l+1}(g) = \sum_{i=0}^{N_l-1} c_{ij}^{l+1}(g) S_{ij}^{l+1}(g) \forall 0 \le j \le N_{l+1} - 1, \forall g \in G$

$f_j^{l+1}(g) = Squash(f_j^{l+1}(g)) = \frac{\|f_j^{l+1}(g)\|_2}{1 + \|f_j^{l+1}(g)\|_2^2} f_j^{l+1}(g); \forall 0 \le j \le N_{l+1} - 1, \forall g \in G$

    **procedure** DYNAMICROUTING($S_{00}^{l+1}(g), ..., S_{N_l-1N_{l+1}-1}^{l+1}(g)$)

$b_{ij}(g) \leftarrow 0; \forall 0 \le i \le N_l - 1, \forall 0 \le j \le N_{l+1} - 1$

for $r$ iterations **do**:

    $c_{ij}(g) = \frac{\exp(b_{ij}(g))}{\sum_{k=0}^{N_{l+1}-1} \exp(b_{ik}(g))}; 0 \le i \le N_l - 1, \forall 0 \le j \le N_{l+1} - 1$

    $f_j^{l+1}(g) = \sum_{i=0}^{N_l-1} c_{ij}^{l+1}(g) S_{ij}^{l+1}(g) \forall 0 \le j \le N_{l+1} - 1$

    $f_j^{l+1}(g) = Squash(f_j^{l+1}(g)) = \frac{\|f_j^{l+1}(g)\|_2}{1 + \|f_j^{l+1}(g)\|_2^2} f_j^{l+1}(g); \forall 0 \le j \le N_{l+1} - 1$

    $b_{ij}(g) = b_{ij}(g) + \langle S_{ij}^{l+1}(g), f_j^{l+1}(g) \rangle; 0 \le i \le N_l - 1, \forall 0 \le j \le N_{l+1} - 1$

    **return** $c_{ij}(g) \forall 0 \le i \le N_l - 1, \forall 0 \le j \le N_{l+1} - 1$

---

activation value exist; for example, spectral routing (Bahadori, 2018) computes the activation score from the sigmoid of the first singular value of the matrix of stacked predictions.

## E  GENERALISATION TO OTHER GROUPS

Our theoretical results and algorithms admit a generalisation to other groups - as long as an appropriate group-convolution is defined. The equivariance and the preservation of detected compositionality is preserved under the condition that the group-convolution is equivariant.

As an example, consider the discrete translation group $\mathbb{Z}^2$ and the regular correlation operation defined for an input with $d$ channels by $(f \star \Psi)(x) = \sum_{t \in \mathbb{Z}^2} \sum_{k=0}^{d-1} f_k(t)\Psi_k(x - t)$. The translation-equivariance of this operation is proved in (Cohen & Welling, 2016).

The general $n$-dimensional correlation defined on $\mathbb{Z}^n$ is given by $(f \star \Psi)(x) = \sum_{t \in \mathbb{Z}^n} \sum_{k=0}^{d-1} f_k(t)\Psi_k(x - t)$. This operation is equivariant to translations in $n$-dimensions. The proof for this is given below.

**Theorem E.1.** *The $n$-dimensional correlation operator is equivariant with respect to $\mathbb{Z}^n$ and the group representation L.*

*Proof.* Consider $x, y, t \in \mathbb{Z}^n$, and $f : \mathbb{Z}^n \to \mathbb{R}^d$. Then, $([L_y f] \star \Psi)(x) = \sum_{t \in \mathbb{Z}^n} \sum_{k=0}^{d-1} f_k(t - y)\Psi_k(x - t) = \sum_{t \in \mathbb{Z}^n} \sum_{k=0}^{d-1} f_k(t)\Psi_k(x - t + y) = \sum_{t \in \mathbb{Z}^n} \sum_{k=0}^{d-1} f_k(t)\Psi_k(x - (t - y)) = [L_y(f \star \Psi)](x)$. Thus, $([L_y f] \star \Psi)(x) = [L_y(f \star \Psi)](x)$. □

Our degree-centrality based algorithm, with its use of discrete convolutions, can be used in its current form with the above convolution. The proof of equivariance and the preservation of compositionality holds from a direct application of the above result to Theorem 2.1 and Theorem 2.2, using the underlying group as $Z^n$.

For continuous groups such as $SO(n)$, the degree-centrality based algorithm must use equivariant convolutions defined over it to remain equivariant. We consider the specific case of $SO(3)$ below.

The correlation of two functions $f, \Psi : SO(3) \to \mathbb{R}^d$ is given by:

$$(f \star \Psi)(R) = \int_{SO(3)} \sum_{k=0}^{d-1} \Psi_k(R^{-1}Q)f_k(Q)dQ. \tag{2}$$

This correlation is equivariant to transformations in $SO(3)$, with respect to the group representation $L_R$ defined by $[L_r f(Q)] = f(R^{-1}Q)$, as proved in (Cohen et al., 2018b). It is to be noted that due to approximations introduced by the sampling of continuous functions in implementations, exact equivariance is not preserved.

However, our routing algorithm can still be used with such convolutions and does not contribute to any reduction of equivariance by itself. This is due to the equivariance of the dot-product and the post-composition operators. The equivariance of the post-composition operator was proved in (Cohen & Welling, 2016). We formally prove the equivariance of dot-product for the $SO(3)$ group.

**Theorem E.2.** *The dot-product between two equivariant functions $f, g : SO(3) \to R^d$ is equivariant with respect to the group representation L. That is, $[L_R f].[L_R g] = L_R[f.g]$*

*Proof.* $[L_R f].[L_R g](Q) = \sum_{k=0}^{d-1} f_k(R^{-1}Q)g_k(R^{-1}Q) = [\sum_{k=0}^{d-1} f_k g_k](R^{-1}Q) = f.g(R^{-1}Q) = L_R[f.g]$. □

The proof for the preservation of compositionality also holds by considering the infinite graph $\mathbb{G}(x)$. The definition for this is the same as before. The proof follows by using the same mapping between vertices, and from the equivariance of the routing procedure.

