# OpenReview forum: "Building Deep Equivariant Capsule Networks"
_ICLR.cc/2020/Conference — Accept (Talk)_

### Official Review · AnonReviewer1 · 2019-10-21
**Official Blind Review #1**

**Rating:** 6

**Review:**

In this work, a method was proposed to train capsule network by projectively encoding the manifold of pose-variations, termed the space-of-variation (SOV), for every capsule-type of each layer. Thereby, the proposed method aims to improve equivariance of capsule nets with respect to translation (rotation and scaling).

The proposed method is interesting and the initial results are promising. However, there are various major and minor problems with the work:

- There are various undefined functions and mathematical notation, such as the following:

- Please give formal and precise definitions of groups and group representations for readers who are not familiar with mathematical groups.

- What are GetWeights and Agreement used in Algorithm 1?

- Please define “routing among capsules” more precisely.

- How do you calculate Pool() more precisely?

- In the paper, the results are given for a general class of groups. However, it is not clear how these results generalize even for some popularly employed groups, such as Z^n, S_n, SO(n), SE(n) etc., with different symmetry properties, base space, and field type.

- Please check the following paper for a detailed discussion on group equivariant CNNs with different group structures, and elaborate the theoretical results for particular groups (e.g. at least for p4m used in experiments) :

T.S. Cohen, M. Geiger, M. Weiler, A General Theory of Equivariant CNNs on Homogeneous Spaces, NeurIPS 2019

- Please define the norms used in Algorithm 2.

- How do you calculate accuracy of models? Are these numbers calculated for a single run, or for an average of multiple runs? If it is the former, please repeat the results for multiple runs, and provide average accuracy with variance/standard deviation. If it is the latter, please provide the variance/standard deviation as well.

- Have you performed analyses using larger datasets such as Cifar 100 or Imagenet? It would be great to provide some results for larger datasets to explore their scalability.

- Please define accuracy given in tables more precisely, use dot "." at the end of sentences in captions.

- There are several typo/grammatical errors, such as the following:

-- Homegenous -> homogeneous

--  for with prediction networks

-- Please proof-read the paper in detail and fix the typo etc.

After the discussions:

Most of my questions were addressed and the paper was improved in the discussion period. Therefore, I increase my rating.

However, some parts of the paper still need to be clarified. For instance;

- GetWeights: To ensure that the predictions are combined in a meaningful manner, different methods can be used to obtain the weights. The role of GetWeights is to represent any such mechanism.

-> Please define these methods in detail and more precisely, at least in the Supp. mat.

- Agreement : The Agreement function represents any means of evaluating such consensus.

-> This is also a very general concept, which should be more precisely defined.

- Our theoretical results (Theorem 2.1 and Theorem 2.2) hold for all general groups, and the particular group representation  defined in the main paper.

-> Could you please give a concrete discussion on generalization of these results and the proposed algorithms for discrete and continues groups? For instance, how do these algorithms and results generalize with Z^n and SO(n)?


**Experience Assessment:**

I have published in this field for several years.

**Review Assessment: Checking Correctness Of Derivations And Theory:**

I assessed the sensibility of the derivations and theory.

**Review Assessment: Checking Correctness Of Experiments:**

I carefully checked the experiments.

**Review Assessment: Thoroughness In Paper Reading:**

I read the paper at least twice and used my best judgement in assessing the paper.

---

> ### Author Response · Authors · 2019-11-11
> **Reply to Reviewer #1**
>
> Thank you for sharing your valuable feedback. Please find our responses to your comments, below. (C = Comment; R = Our Response). We have modified the paper to reflect your review, and are very interested in any further feedback from you.
> ...........................................................................................................
> C: There are various undefined functions and mathematical notation, such as the following:
>    - Please give formal and precise definitions of groups and group representations for readers who are not familiar with mathematical groups.
>    - What are GetWeights and Agreement used in Algorithm 1?
>    - Please define “routing among capsules” more precisely.
>    - How do you calculate Pool() more precisely?
>
> R: We have updated the paper to include more precise definitions for these. Specifically, we have done the following:
> 1. Added the formal definitions for a group and a group representation to the appendix A.1 and A.2.
> 2. We realise that we have used the term “routing among capsules” in section 2 without reference to its description in section 2.1. We have modified the paper to refer to this subsection. It is to be noted that a general description of routing was given in section 1, paragraph 3.
> 3. We have added a more formal definition of Pool() in the appendix A.3, due to space constraints.
> 4. We have added descriptions for the roles of the "GetWeights" and "Agreement" functions in section 2.1.
>
> We also include extended explanations for some of these modifications in this comment, as given below.
>
> 1. Definition of group, group action, and group representation
>
> Formal definition of a group: A tuple (G, .), where G is a non-empty set and . defines a binary operation on G, is said to form a group if the following properties are satisfied:
>
> a. Closure: For all g1, g2 in G, g1.g2 belongs to G.
> b. Associativity: For all g1, g2, g3 in G, (g1.g2).g3 = g1.(g2.g3)
> c. Existence of the identity element: There exists e in G, such that for all g in G, e.g = g.e = g.
> d. Existence of an inverse: For each g in G, there exists g^{-1} in G, such that g.g^{-1} = g^{-1}.g = e.
>
> Formal definition of a group action and group representation: Given a group (G,.) and a vector space V, a group action is a function f from G x V to V satisfying the following properties.
>
> a. For all a in V, f(e, a) = a.
> b. For all g, h in G and for all a in V, = f(h, f(g, a)) = f(h.g, a).
>
> A group representation is a group action by invertible linear maps. More formally, a group representation of a group (G, .), with respect to a vector space V, is a homomorphism from G to GL(V) - the set of linear, invertible maps from V to V.
>
> 3. Definition of Pool(g)
> Consider a one-layer GCNN-convolutional prediction network $\Psi_{j}^{l+1}$ for a SOVNET layer $l+1$, and for the $d^{l+1}$-dimensional $j^{th}$ capsule-type. Intuitively, $Pool_{j}^{l+1}(g)$ is defined by the extent of the support of the g-transformed filter. More formally, $Pool_{j}^{l+1}(g)$ = $\{h \in G: \Psi_{j}^{l+1}(g^{-1}\circ h) \neq 0\}$.
> For a general $L$-layer GCNN prediction- network, $Pool_{j}^{l+1}(g)$  is defined by recursively applying the above definition through all the layers of the prediction network.
>
> 4. Description of GetWeights and Agreement
> The weighted-sum family of routing algorithms described in Algorithm 1 build deeper capsules using a weighted sum of predictions made for them by shallower capsules. To ensure that the predictions are combined in a meaningful manner, different methods can be used to obtain the weights. The role of GetWeights is to represent any such mechanism.
>
> The activation of a capsule, representative of the probability of existence of the object it represents, is determined by the extent of the consensus among its predictions. This is based on the routing-by-agreement principle of capsule networks. The Agreement function represents any means of evaluating such consensus.
> ............................................................................................................
>
> C: Please define the norms used in Algorithm 2.
>
> R: The norm used in our model is the vector 2-norm, given by $\Vert x \Vert_{2}$ = $(\sum_{i=0}^{n-1}x_{i}^{2})^{\frac{1}{2}}$, where $x$ $\in \mathbb{R}^{n}$. We have added this information to the revised paper.
> ............................................................................
>
> C: - Please define accuracy given in tables more precisely, use dot "." at the end of sentences in captions.
>
> R: We define accuracy as the number of correctly classified test-instances divided by the total number of test-instances. The accuracies given in the transformation-robustness experiments (Tables 2 to 4, section 3) are obtained for each dataset and model, by first training a model with a transformed version of a train dataset, and then reporting this accuracy for several transformed versions of the test dataset.
>
>  Further we have modified the captions as pointed out.

---

> > ### Author Response · Authors · 2019-11-11
> > **Continuation of reply to Reviewer #1**
> >
> > C: - Have you performed analyses using larger datasets such as Cifar 100 or Imagenet? It would be great to provide some results for larger datasets to explore their scalability.
> >
> > R: We have trained a SOVNET architecture on CIFAR100. Our model has achieved an accuracy of 71.55%, an almost 4 percentage improvement over a recent capsule network model - STARCAPS [5] which achieved 67.66%. We have reported this result in the appendix C.4. The code for this is available in the associated github repository.
> >
> > We wish to thank you for this insightful comment, but also would like to point out that testing on large scale datasets is not (yet) commonplace for capsule networs, as the field is still in its infancy. In fact, none of the baseline capsule network models that we considered showcased results on CIFAR100. Moreover, we would also like to point out the significant experimentation we have performed on MNIST, FashionMNIST, and CIFAR10, as well on KMNIST and SVHN. To the best of our knowledge, ours is the most comprehensive capsule-network based study on transformation robustness on a variety of datasets. We sincerely hope that the contribution of these results is not undermined or ignored.
> > .........................................................................................................
> >
> > C: - There are several typo/grammatical errors, such as the following:
> >       -- Homegenous -> homogeneous
> >       --  for with prediction networks
> >       -- Please proof-read the paper in detail and fix the typo etc.
> >
> > R: Please excuse us for our typographical errors, we have fixed them.
> > .........................................................................................................
> >
> > C: - How do you calculate accuracy of models? Are these numbers calculated for a single run, or for an average of multiple runs? If it is the former, please repeat the results for multiple runs, and provide average accuracy with variance/standard deviation. If it is the latter, please provide the variance/standard deviation as well.
> >
> > R: - Due to the large number of experiments - over 300 values tabulated ( Table 2 to Table 5, section 3) - we found it infeasible to perform several runs for the baselines as well as our architecture. However, we did train and test our SOVNET architectures multiple times, and found that the results are consistent, though the numbers are not recorded.

---

> > > ### Author Response · Authors · 2019-11-11
> > > **Continuation of reply to Reviewer #1**
> > >
> > > C: - In the paper, the results are given for a general class of groups. However, it is not clear how these results generalize even for some popularly employed groups, such as Z^n, S_n, SO(n), SE(n) etc., with different symmetry properties, base space, and field type.
> > >
> > > - Please check the following paper for a detailed discussion on group equivariant CNNs with different group structures, and elaborate the theoretical results for particular groups (e.g. at least for p4m used in experiments) :
> > >
> > > T.S. Cohen, M. Geiger, M. Weiler, A General Theory of Equivariant CNNs on Homogeneous Spaces, NeurIPS 2019
> > >
> > > R: Our theoretical results (Theorem 2.1 and Theorem 2.2) hold for all general groups, and the particular group representation $L_{g}$ defined in the main paper. Thus, they are true for groups such as Z^n, S_n, SO(n), SE(n) etc. In other words, there is no specific group-dependence of our results - as long as we are able to define a group-convolution.
> > >
> > > The scope of our paper and that of Cohen et. al [6] are very different. Their paper aims to build a general theory of equivariant CNNs, and they show that convolutions with equivariant kernels are the most general class of equivariant maps between feature spaces. It is to be noted that they do not report any empirical results at all.
> > >
> > > Whereas our paper describes a means of integrating equivariant convolutions with capsule networks, thus lending equivariance guarantees to capsule networks - something that is, in general, lacking in the field of capsule networks. To supplement the theory in our work, we also perform several experiments (more than 75 experiments with over 300 values tabulated).
> > >
> > > Our model fits into the framework of Cohen et.al [6]; however, a description or a detailed discussion of their paper in our work is not in line with our goals, and also given the limitations in space. We have, however, made a mention of this paper in section 4 – where related work is discussed - and have included it in our references as it is an important paper in the literature pertaining to equivariant convolutions.
> > >
> > > We quote from [7] - the arxiv version of [6] - the total space, the base space, the stabiliser, and the category of representation for p4 and p4m convolutions:
> > >
> > > Total Space     Stabiliser    Base space    Category of representation
> > >
> > > p4                        C4              Z^{2}              Regular
> > > p4m                    D4              Z^{2}              Regular
> > >
> > > Where C4 is the cyclic group of order 4 (here it corresponds to the 4 multiples of 90 degree rotations)
> > >       D4 is the dihedral group of order 8 (here it corresponds to all possible compositions of two mirror reflections and the 4 multiples of 90 degree rotations)
> > >
> > > ..........................................................................................................
> > >
> > > References
> > > 1. Sabour, Sara, Nicholas Frosst, and Geoffrey E. Hinton. "Dynamic routing between capsules." Advances in neural information processing systems, 2017.
> > > 2. Hinton, Geoffrey E., Sara Sabour, and Nicholas Frosst. "Matrix capsules with EM routing." ICLR, 2018.
> > > 3. Bahadori, Mohammad Taha. "Spectral capsule networks." ICLR, 2018.
> > > 4. Wang, Dilin, and Qiang Liu. "An optimization view on dynamic routing between capsules." ICLR, 2018.
> > > 5. Karim Ahmed, Lorenzo Torresani. "Star-Caps: Capsule Networks with Straight-Through Attentive Routing." NeurIPS, 2019.
> > > 6. T.S. Cohen, M. Geiger, M. Weiler, "A General Theory of Equivariant CNNs on Homogeneous Spaces.", NeurIPS, 2019.
> > > 7. Cohen, Taco, Mario Geiger, and Maurice Weiler. "A general theory of equivariant cnns on homogeneous spaces." arXiv preprint arXiv:1811.02017, 2018.

---

> ### Author Response · Authors · 2019-11-15
> **Reply to additional comments**
>
> We thank you for upgrading the score of our paper.
>
> We have added a discussion based on your comments to appendix D and E, respectively. Briefly, it summarised as follows:
>
> We provided examples of GetWeights and Agreement so as to clarify their role. Please note that a formalisation of these two concepts to cover a general case of consensus-based importance is not done. Such a formalisation does not currently exist, to the best of our knowledge. We hope the role of these functions is clarified by the examples we provide.
>
> We discussed the case of Z^{n} and SO(3), as examples of discrete and continuous groups which can be used with our algorithm. Our algorithm preserves equivariance conditioned on the use of appropriate convolutions. Thus, our theoretical results hold. However, practical implementations of convolutions on continuous groups involve sampling that leads to loss of exact equivariance. Thus, while our routing algorithm preserves equivariance, sampling of continuous functions for implementation of convolutions results in a loss of this property.

---

### Official Review · AnonReviewer2 · 2019-10-23
**Official Blind Review #2**

**Rating:** 8

**Review:**

This paper combines CapsuleNetworks and GCNNs with a novel formulation. First they modify the CapsNet formulation by replacing the linear transformation between two capsule layers with a group convolution. Second they share the group equivarient convolution filters per all capsules of the lower layer.  Third, they change the similarity metric from a lower-upper similarity into a pairwise lower similarity and aggregation which makes it keep the equivarience. Since the cij does not depend on upper capsule anymore they only perform 1 routing iteration (no modification of the routing factors).

One assumption in CapsNets is that each part belongs to one whole. Therefore, the normalization in Alg.2 usually is division by degree^k_i. The proposed normalization formula for c_ij seems to encourage that each upper capsule only receives one part. Is this a typo or is there a justification for this?

The discussion on ideal graph on page 5 is interesting. But the points made are not used later on. I expected the results to have an analysis or at least a show case that indeed if you transform the resultant graphs stay isomorphic.

One goal for CapsuleNetworks vs GCNNs is the hope for handling different transformations and not only rotations that one can grid with group convolutions. But, the experiments only report on rotation, translation as a transformation. Reporting results by training on MNIST, testing on AFFNIST could shed light on this aspect of SOVNETs.

Conditioned that the last two points will be addressed in the rebuttal I vote for accepting this paper since they suggest a novel formulation that brings some measures of rotation equivarience guarantee into CapsNets. Also their results suggest that there is no need for per Capsule filter bank and several refinements to get rotation robustness (it would be interesting to check the performance of a simple capsnet with shared parameters).  In the appendix there is a comparison with GCNNs on fashion MNIST which shows they have better performance than GCNNs. I would advise reporting GCNNs for all the experiments in the main paper.


------------------------------------------
Thank you for updating and expanding the paper. The extra experiments, isomorphism analysis and their response regarding the attention vs part-whole makes the paper much stronger. Therefore, I am increasing my score.

**Experience Assessment:**

I have published in this field for several years.

**Review Assessment: Checking Correctness Of Derivations And Theory:**

I assessed the sensibility of the derivations and theory.

**Review Assessment: Checking Correctness Of Experiments:**

I carefully checked the experiments.

**Review Assessment: Thoroughness In Paper Reading:**

I read the paper thoroughly.

---

> ### Author Response · Authors · 2019-11-10
> **Reply to Reviewer #2**
>
> Reply to Reviewer 2
>
> Thank you for sharing your valuable feedback, and acknowledging our contributions. Please find our responses to your comments, below. (C = Comment; R = Our Response). We will modify the paper to reflect your review, and are very interested in any further feedback from you.
>
> edit: We have modified the paper, and uploaded the codes. The results of the GCNN experiment for CIFAR10 will be added in a couple of days.
> .......................................................................................................................................................................................................
>
> C: One assumption in CapsNets is that each part belongs to one whole. Therefore, the normalization in Alg.2 usually is division by degree^k_i. The proposed normalization formula for c_ij seems to encourage that each upper capsule only receives one part. Is this a typo or is there a justification for this?
>
> R: The normalisation scheme used in our degree-routing algorithm is intentional, and not a typo. Our intuition and explanation for this follows.
>
> Each routing algorithm, at least in the weighted-sum family, defines a means of evaluating the relative strengths of connections between lower and upper-capsules. These are given quantitatively by the routing weights. We emphasise that the means of normalisation of these weights must be seen in the context of the method used to obtain them.
>
> In methods that normalise among upper-capsules, such as "dynamic routing" by Sabour at. al [1], the un-normalised weights denote the similarity between a prediction for an upper-capsule, and an intermediate vector-value of that capsule. Thus, in such methods, normalisation of weights among upper-capsules, given a fixed lower-capsule, models the relative importance amongst upper-capsules for that lower-capsule. The upper-capsule with the largest similarity to the prediction made for it by the fixed lower-capsule gets the maximum normalised weight. Thus, in scenarios where routing-weights model the "attention" that lower capsules give to upper-capsules, normalisation among the latter is meaningful.
>
> This is in contrast to our degree-routing procedure. We aim to capture and use consensus among predictions for a fixed upper-capsule so as to build agreement-based, rather than attention-based, upper-capsules. Thus, the main aim is to give larger weights to predictions (for a fixed upper-capsule) that exhibit greater consensus with respect to their peers. Thus, it is entirely possible that two predictions are close in their overall consensus behaviour, and would have similar weights, causing multiple parts to route to a single whole. One means of assigning such weights, is to consider the degree scores for each prediction (treating the predictions as being vertices of a similarity-weighted, complete graph). By using these scores in a weighted summation, we aim to build a deeper capsule keeping in mind the principle of routing-by-agreement as espoused in the paper "Transforming autoencoders" by Hinton et al.[2]. The normalisation among lower-capsules is merely a means to ensure that the weights are in the range (0,1).
>
> It is to be noted that normalising across the upper-capsules instead of lower-capsules in our method would lead to comparison of degree-scores of different prediction-graphs, and would not be meaningful.

---

> > ### Author Response · Authors · 2019-11-10
> > **Continuation of reply to reviewer #2**
> >
> > C: The discussion on ideal graph on page 5 is interesting. But the points made are not used later on. I expected the results to have an analysis or at least a show case that indeed if you transform the resultant graphs stay isomorphic.
> >
> > R: We have performed two experiments to verify that the capsule decomposition-graphs of the transformed and untransformed images are isomorphic.
> >
> > For the first of these, we trained a P4 convolution based SOVNET architecture on untransformed images of MNIST. We then considered four variations of the MNIST test-dataset - untransformed, and three versions rotated exactly by multiples of 90 degrees: 90, 180, and 270. Our experiment verifies that the mapping defined in the proof of Theorem 2 (given in the appendix, page 13, Theorem A.2) is indeed an isomorphism.
> >
> > To this end, we considered the capsule-activations as well as the degree-scores, obtained across all the capsule-layers, for each image of all the variations of the test split. We then mapped the activations and the degree-scores for the untransformed images by the aforesaid mapping for each of the transformations. This corresponds to 'rotating' the activations and degree-scores by each transformation. We then computed the squared error of these with each of the activations and degree-scores obtained from the correspondingly transformed image, respectively. A successful verification would result in zero error (up to machine precision).
> >
> > The results below show that this happens.
> >
> >     Rotation     Mean-squared error for capsule-activations    Mean-squared error for degree-scores
> >        90             6.1900e-15                                                                3.3087e-15
> >       180            6.2821e-15                                                                3.3606e-15
> >       270            6.1911e-15                                                                3.3138e-15
> >
> > The second of our experiments is an empirical verification that the test-accuracies remain unchanged under transformations for which SOVNET exhibits equivariance. We use the same trained architecture as above, and verify that the accuracy remains unchanged under exact transformations of the images. We present the results below.
> >
> >    Rotation       Accuracy
> >      0                 99.52%
> >     90                99.52%
> >    180               99.52%
> >    270               99.52%
> >
> > We repeated the same experiments for FashionMNIST - the results are presented below. Results for CIFAR-10 will be updated shortly. We emphasize that the accuracies reported for these experiments are the result of simple architectures, whose primary aim is to verify theorem 2, and not any limitation of the SOVNET model.
> >
> >     Rotation     Mean-squared error for capsule-activations     Mean-squared error for degree-scores
> >        90             2.5678e-13                                                                   1.9576e-13
> >       180            2.6306e-13                                                                   1.9981e-13
> >       270            2.5869e-13                                                                   1.9662e-13
> >
> >     Rotation       Accuracy
> >        0                92.23%
> >       90               92.23%
> >      180              92.23%
> >      270              92.23%
> >
> > We note that the architecture used for these experiments does not use residual blocks during the initial convolution stage and in the prediction networks. The main reason for this is that strided convolution layers (used in residual blocks) cause a loss in provable equivariance. Thus, we use only unstrided, simple convolutions for these experiments.
> >
> > However, these architectural differences aside, our architectures are still within the framework of the SOVNET model. They use P4 group-equivariant convolutional prediction-mechanisms, and degree-routing. The use of strided residual-blocks in the previous experiments was to have a mix of equivariant networks for transformation-robustness and residual-connections for better performance.

---

> > > ### Author Response · Authors · 2019-11-10
> > > **Continuation of reply to Reviewer #2**
> > >
> > > C: One goal for CapsuleNetworks vs GCNNs is the hope for handling different transformations and not only rotations that one can grid with group convolutions. But, the experiments only report on rotation, translation as a transformation. Reporting results by training on MNIST, testing on AFFNIST could shed light on this aspect of SOVNETs.
> > >
> > > R: We trained a SOVNET architecture on MNIST images that are padded to size 40x40 - the size of AFFNIST images. We augment these images by translations in the range of [-6,6] pixels, as is the standard approach. Note that the changed size of the images necessitates a different architecture. The result of this experiment is given below. We see that our SOVNET architecture obtains the highest accuracy when compared to other recent capsule network models.
> > >
> > >       Model                                Accuracy
> > > Sabour et. al [1]                     79.0%
> > > Hinton et. al [3]                      93.1%
> > > Lenssen et. al [4]                   89.10%
> > > Jeong et. al [5]                        87.8%
> > > Choi et. al [6]                          91.6%
> > > SOVNET (only translation)   97.91%
> > >
> > >
> > > We also trained the above SOVNET architecture on MNIST with translations in the range of [-6,6] pixels and rotations from [-30 ,30] degrees. While this increases the extent of train-time augmentation, there are several test-time transformations that are unseen.  With this scheme, we achieve state-of-the-art accuracy of 99.20%. This improves over the best, to our knowledge, accuracy of 98.3% obtained by Tai et. al [7]
> > >
> > >
> > > The performance of the SOVNET architecture showcases its ability to generalise to unseen affine transformations.
> > > ............................................................................................................
> > >
> > > C:  In the appendix there is a comparison with GCNNs on fashion MNIST which shows they have better performance than GCNNs. I would advise reporting GCNNs for all the experiments in the main paper.
> > >
> > > R: THe complete set of results on MNIST and FashionMNIST for the transformation-robustness experiments as in the main text have been given below. The results for the experiments on CIFAR10 will be given shortly. All the values are accuracies in percentage.
> > >
> > >                                   Experiments on MNIST
> > >            Trained on (0,0)                                             Trained on (2,30)
> > >                 (0,0)  (2,30) (2,60) (2,90) (2,180)           (0,0)  (2,30)  (2,60) (2,90) (2,180)
> > > GCNN     99.61, 93.96, 75.53, 58.91, 46.07           99.67,  99.46, 97.11, 84.5,  63.74
> > > SOVNET 99.68, 96.15, 80.53, 64.55, 51.02          99.77,  99.70, 98.86, 90.63, 69.26
> > >
> > >
> > >                 Trained on (2,60)                                             Trained on (2,90)
> > >                 (0,0)  (2,30) (2,60) (2,90)  (2,180)                   (0,0)   (2,30)  (2,60)  (2,90) (2,180)
> > > GCNN     99.52, 99.38, 99.37, 97.02,  74.98                   89.34,  89.16,  89.13,  88.86, 75.53
> > > SOVNET 99.70, 99.65, 99.63  98.56   79.59                   99.68,  99.60,  99.59,  99.5,  87.76
> > >
> > >                  Trained on (2,180)
> > >        (0,0)   (2,30)  (2,60) (2,90) (2,180)
> > >       GCNN   87.8,   87.51,  87.47, 87.41, 87.45
> > >       SOVNET 98.34,  98.10,  98.11, 98.08, 98.06
> > >
> > >                                   Experiments on FashionMNIST
> > >                    Trained on (0,0)                                Trained on (2,30)
> > >                   (0,0)  (2,30) (2,60) (2,90)  (2,180)     (0,0)  (2,30) (2,60) (2,90) (2,180)
> > > GCNN       84.63, 56.23, 37.31, 0.2862, 21.58    92.25, 90.95, 72.17, 51.93, 37.12
> > > SOVNET   94.72, 61.58, 41.01, 34.07,  27.63     94.99, 94.36, 77.19, 58.59, 43.84
> > >
> > >                   Trained on (2,60)                       Trained on (2,90)
> > >                   (0,0)  (2,30) (2,60) (2,90) (2,180)       (0,0)  (2,30) (2,60) (2,90) (2,180)
> > > GCNN       90.78, 89.82, 89.67, 76.69, 49.97       90.31, 89.46, 89.42, 89.22, 64.44
> > > SOVNET  94.49, 94.08, 94.20, 90.23, 73.48       94.41, 94.03, 93.93, 93.98, 91.42
> > >
> > >                   Trained on (2,180)
> > >                   (0,0)  (2,30) (2,60) (2,90) (2,180)
> > > GCNN       89.7,  88.65, 88.61, 88.62, 88.6
> > > SOVNET  94.11, 93.77, 93.56, 93.57, 93.60
> > >
> > >
> > > As can be seen in the tables, the SOVNET architecture performs better than the GCNN architecture in all the cases. We will add these results in the main text of the paper.
> > >
> > > While we would very much like to include the other aforementioned results in the main text of the paper, due to space constraints, we will add them in the appendix.
> > > ............................................................................................................
> > > edit: we have updated the accuracy on affnist based on the results of our model.
> > >
> > > The code for all of these experiments will be made available in the associated github repository https://github.com/AnonymousCapsuleSOVNET/SOVNET within a day.

---

> > > > ### Author Response · Authors · 2019-11-10
> > > > **Continuation of reply to Reviewer #2**
> > > >
> > > > References
> > > > 1. Sabour, Sara, Nicholas Frosst, and Geoffrey E. Hinton. "Dynamic routing between capsules." Advances in neural information processing systems. 2017.
> > > > 2. Hinton, Geoffrey E., Alex Krizhevsky, and Sida D. Wang. "Transforming auto-encoders." International Conference on Artificial Neural Networks. Springer, Berlin, Heidelberg, 2011.
> > > > 3. Hinton, Geoffrey E., Sara Sabour, and Nicholas Frosst. "Matrix capsules with EM routing." ICLR, 2018.
> > > > 4. Lenssen, Jan Eric, Matthias Fey, and Pascal Libuschewski. "Group equivariant capsule networks." Advances in Neural Information Processing Systems. 2018.
> > > > 5. Jeong, Taewon, Youngmin Lee, and Heeyoung Kim. "Ladder Capsule Network." International Conference on Machine Learning. 2019.
> > > > 6. Choi, Jaewoong, et al. "Attention routing between capsules." Proceedings of the IEEE International Conference on Computer Vision Workshops. 2019.
> > > > 7. Tai, Kai Sheng, Peter Bailis, and Gregory Valiant. "Equivariant Transformer Networks." arXiv preprint arXiv:1901.11399 (2019).

---

> ### Author Response · Authors · 2019-11-15
> **newest affnist results**
>
> The latest results on affnist have been added to the response below and the paper in appendix C, Table 8.

---

### Author Response · Authors · 2019-11-13
**Summary of responses to reviewers**

We thank the reviewers for their insightful comments. We have revised the paper and responded to their queries. We summarise the major changes to our paper below.

Summary of major changes to our paper

1. We have added missing definitions for group, group representation, and Pool in the appendix A as required by reviewer1.
2. We added description for GetWeights and Agreement functions in section 2; made the reference to 'routing among capsules' more clear in section 2 as required by reviewer 1.
3. We specified the norm used in the paper as required by reviewer 1.
4. We added results of SOVNET on CIFAR100 to the appendix C as required by reviewer 1.
5. We added the paper [1] to our references and related work in response to reviewer 1's comment.
6. We added results of experiments to verify the isomorphism of the capsule graph-decomposition to the appendix C as required by reviewer 2.
7. We added results of experiments on AFFNIST to the appendix C as required by reviewer 2.
8. We added the results of the transformation-robustness experiments for the group-convolutional networks to the tables 2 to 4 as required by reviewer 2.
9. We added the results of a simple capsnet model with shared parameters on MNIST and FashionMNIST to the appendix C in response to reviewer 2's comment .

We also fixed typos in our paper.

The codes for the experiments have been uploaded to our anonymous repository.

References
[1] T.S. Cohen, M. Geiger, M. Weiler, "A General Theory of Equivariant CNNs on Homogeneous Spaces.", NeurIPS, 2019.

---

### Decision · Program_Chairs · 2019-12-19

**Decision:**

Accept (Talk)

**Comment:**

This paper combine recent ideas from capsule networks and group-equivariant neural networks to form equivariant capsules, which is a great idea. The exposition is clear and the experiments provide a very interesting analysis and results. I believe this work will be very well received by the ICLR community.